# Developmental, Physiological and Phylogenetic Perspectives on the Expression and Regulation of Myosin Heavy Chains in Craniofacial Muscles

**DOI:** 10.3390/ijms25084546

**Published:** 2024-04-21

**Authors:** Joseph Foon Yoong Hoh

**Affiliations:** Discipline of Physiology, School of Medical Sciences, Faculty of Medicine and Health, The University of Sydney, Sydney, NSW 2006, Australia; joeh@iinet.net.au

**Keywords:** craniofacial muscles, myosin heavy chain, gene expression, myogenesis, neural regulation, thyroid hormone, muscle plasticity, phylogeny

## Abstract

This review deals with the developmental origins of extraocular, jaw and laryngeal muscles, the expression, regulation and functional significance of sarcomeric myosin heavy chains (MyHCs) that they express and changes in MyHC expression during phylogeny. Myogenic progenitors from the mesoderm in the prechordal plate and branchial arches specify craniofacial muscle allotypes with different repertoires for MyHC expression. To cope with very complex eye movements, extraocular muscles (EOMs) express 11 MyHCs, ranging from the superfast extraocular MyHC to the slowest, non-muscle MyHC IIB (nmMyH IIB). They have distinct global and orbital layers, singly- and multiply-innervated fibres, longitudinal MyHC variations, and palisade endings that mediate axon reflexes. Jaw-closing muscles express the high-force masticatory MyHC and cardiac or limb MyHCs depending on the appropriateness for the acquisition and mastication of food. Laryngeal muscles express extraocular and limb muscle MyHCs but shift toward expressing slower MyHCs in large animals. During postnatal development, MyHC expression of craniofacial muscles is subject to neural and hormonal modulation. The primary and secondary myotubes of developing EOMs are postulated to induce, via different retrogradely transported neurotrophins, the rich diversity of neural impulse patterns that regulate the specific MyHCs that they express. Thyroid hormone shifts MyHC 2A toward 2B in jaw muscles, laryngeal muscles and possibly extraocular muscles. This review highlights the fact that the pattern of myosin expression in mammalian craniofacial muscles is principally influenced by the complex interplay of cell lineages, neural impulse patterns, thyroid and other hormones, functional demands and body mass. In these respects, craniofacial muscles are similar to limb muscles, but they differ radically in the types of cell lineage and the nature of their functional demands.

## 1. Introduction

Muscles that move different parts of the body are associated with unique properties that equip them to meet different types of functional demands made upon them. The main functions of limb and trunk muscles are locomotion and the maintenance of posture which are well served by four types of muscle fibres, the β-slow and three subtypes of fast fibres, 2a, 2x and 2b. The functional demands on craniofacial muscles are much more varied and complex. Among other functions, they must cope with the complex movements of the eyes, move the jaw in different animals in diverse modes of predation and feeding, regulate respiration and protect the airway. To meet these functional demands, a wider range of muscle fibre types which include those with contractile properties faster and slower than those in limb muscles is needed. The origin of these differences in fibre-type composition between craniofacial and limb muscles and between different craniofacial muscles can be traced to the diversity of the premyogenic mesodermal cell lineages that give rise to them.

### 1.1. Phylogeny and Developmental Origins of Craniofacial Muscles

This review deals with three groups of craniofacial muscles, the extraocular, jaw and laryngeal muscles in mammals. These muscles evolved in different eras during phylogeny. Extraocular muscles (EOMs) are the oldest, being present in jawless fish with an evolutionary history of about 550 million years. The EOMs of the lamprey, a living jawless fish, have four recti and two oblique muscles that are not much different from those in tetrapods. The number and innervation of EOMs of cartilaginous fish and mammals are essentially the same [1,2]. The jaw appeared in cartilaginous fish about 450 million years ago. Jaw-closing muscles are powered by masticatory myosin which is expressed in the living shark [3], and myosin expression in jaw-closing muscles undergoes radical changes during mammalian evolution. Laryngeal muscles only appeared with adaptations of vertebrates to living on land, with air breathing and with the evolution of lungs.

Myogenic cells forming muscles in different regions of the body arise from myogenic progenitor cells of the paraxial mesoderm of the early embryo. These progenitor cells are committed to different lineages of myogenic cells by diverse regulatory factors [4]. Limb and trunk muscles develop from cells in the segmented somites on each side of the developing spinal cord. EOMs develop from cells that originate from the prechordal plate [5,6] and the first branchial arch [7]. Other craniofacial muscles originate from progenitor myoblasts in the unsegmented cranial paraxial mesoderm and from the most rostral occipital somites [8]. These progenitor myoblasts migrate into the branchial arches. Jaw muscle cells are derived from the first branchial arch, while cells of the intrinsic laryngeal muscles originate from the fourth arch and the sixth arch [9]. The expression of the pituitary homeobox gene 2 (Pitx2) commits progenitor myoblasts from the prechordal plate to the EOM myoblast lineage, while the expression of Pitx2 and the T-box gene 1 (Tbx1) commits progenitor myoblasts to the branchial myoblast lineage, in contrast to the expression of the Pitx3 gene which commits somitic progenitors to trunk and limb myoblast lineage [4,10]. Table 1 summarizes the progenitor myoblasts of extraocular, jaw and laryngeal muscles and the transcription factors involved in the specification of the various types of muscles and their nerve supplies. Progenitor myoblasts in the branchial arches and somites are further specified by different transcription factors in the various myogenic lineages and ultimately bring about myogenesis through the four myogenic regulatory factors: Myf5, Mrf4, MyoD and myogenin. Myf5, Mrf4 and MyoD commit progenitor myoblasts to the myogenic fate, while myogenin, the terminal differentiation factor, initiates the differentiation of myoblasts into muscle fibres [11,12].

### 1.2. Myosin Heavy-Chain and Light-Chain Isoforms and Their Genes

Myosin controls the kinetics of chemo-mechanical energy transduction from ATP in muscle and thus the kinetic properties of muscle fibres. The speed of contraction of a muscle is generally proportional to the ATPase activity of its myosin [13]. As the maximal stress of muscles of different speeds is about the same, muscle power (force x velocity) is also a function of myosin ATPase activity. Thus, the myosin isoform expressed in a muscle fibre determines its speed and power and serves as the primary basis for fibre-type classification.

Myosin is a hexamer consisting of a pair of heavy chains (MyHCs) and two pairs of light chains (MLCs) which together form a two-headed molecule with an elongated rod-like tail. The C-terminal halves of the MyHCs are α-helical and wrap around each other to form the tail, while the N-terminal halves form the globular heads, each of which contains an ATPase site and an actin-binding site. Each head is attached to the rod by an α-helical lever arm to which the essential light chain (MLC1) and the regulatory light chain (MLC2) are attached. In striated muscles, myosin molecules assemble into thick filaments through the antiparallel packing of the rod portion of myosin molecules, forming bipolar structures with a central bare zone on either side of which the myosin heads are helically arranged on the filament surface. During muscle contraction, myosin heads act as cross-bridges between thick and thin filaments that cyclically interact with actin molecules of the thin filaments, hydrolyzing ATP and generating force and movement between filaments. During the chemo-mechanical coupling, the small conformational change in the catalytic domain of the head is amplified by the lever arm [14]. This cyclic activity generates the relative force and motion between the thick and thin filaments which underlie the sliding filament theory of muscle contraction [15].

Twelve MyHC genes are known to be expressed in mammalian striated muscles, namely, the eleven striated muscle MyHC genes [16] and the non-muscle MyHC IIB (nmMyH IIB) gene (*MYH10*) which is only expressed in sarcomeres of EOM [17]. The eleven striated muscle MyHC genes include the six skeletal MyHCs which are clustered in tandem in the mammalian genome [18] in the following order: embryonic (*MYH3*), fast 2A (*MYH2*), 2X (*MYH1*), 2B (*MYH4*), neonatal (*MYH8*) and extraocular (EO) (*MYH13*). There are two cardiac MyHC genes, α-cardiac (*MYH6*) and β-cardiac, the latter of which is identical to that of the slow skeletal MyHC gene and is designated as β-slow (*MYH7*). The α-cardiac and β-slow genes are also arranged in tandem in the genome [19,20]. There are two EOM-specific MyHC genes, the slow tonic (*MYH7b*) and slow B (*MYH15*) [21,22], and finally a highly jaw-specific and most ancient masticatory MyHC gene (*MYH16*) [23]. The expression of different MyHCs with their various associated isoforms of MLC1 and MLC2 in different fibres generates muscle fibre types with a wide range of kinetic characteristics to meet the diverse functional demands on craniofacial muscles.

Myosin light chains are classified into those associated with fast muscle fibres (MLC1_F_/MLC3_F_ and MLC2_F_), slow skeletal fibres/ventricular myocardium (MLC1_S_ and MLC2_S_), atrial myocardium (MLC1_A_ and MLC2_A_) and masticatory fibres (MLC1_M_ and MLC2_M_). The essential light chain of fast myosins exists as two isoforms (MLC1_F_ and MLC3_F_) [24] which are spliced variants of the MLC1_F_/MLC3_F_ gene (*MYL1*) [25]. These isoforms share a C-terminal sequence, but differ in the N-terminal region, with MLC1_F_ having 38 amino acid residues more than MLC3_F_ [25].

The functional significance of fast MLC1 heterogeneity is the modulation of cross-bridge kinetics of fast myosins. Fast myosin with MLC3_F_ moves actin filaments faster in motility assays than fast myosin with MLC1_F_ [26]. The maximal velocity of shortening (Vmax) of rabbit fast fibres is proportional to the content of MLC3_F_ [27]. The MLC1_F_ isoform with the larger N-terminal domain has a greater affinity for actin [28], thereby retarding the cross-bridge detachment rate during the cross-bridge cycle, leading to a reduced Vmax.

The MLC1 of embryonic (MLC1_E_) [29], atrial (MLC1_A_) [30] and masticatory (MLC1_M_) [31] myosins were initially thought to be distinct entities. However, later work revealed that MLC1_E_ and MLC1_A_ are identical [25,32] and have been designated as MLC1_E/A_, and that MLC1_M_ is identical to MLC1_E/A_ [33]. MLC1_E/A_ and MLC1_S_ are encoded by *MYL4* and *MYL3,* respectively [34]. Both of these MLC1 isoforms can associate with β-slow MyHC. The N-terminal of MLC1_E/A_ has fewer charged amino acid residues compared with slow/ventricular MLC1_S_ and has a weaker interaction with actin [35]. Consequently, the cross-bridge cycling rate of β-slow myosin with MLC1_E/A_ in cat jaw slow fibres (see later) is considerably higher than that of β-slow myosin with MLC1_S_ in β-slow limb fibres [36].

The regulatory light chain MLC2 in striated muscles exists as four isoforms, each encoded by a distinct gene [37]: the fast skeletal isoform MLC2_F_ encoded by *MYL2_F_* [38], the ventricular/slow skeletal isoform MLC2_S_ by *MYL2* [39,40], the atrial isoform MLC2_A_ by *MYL2_A_* [41] and embryonic/masticatory isoform MLC2_M_ by *MYL5* [42,43].

The functional significance of MLC2 lies in the fact that it helps to stabilize the myosin interacting head motif (IHM, see later), the destabilization of which by MLC2_F_ phosphorylation is the basis of post-tetanic twitch potentiation in which thick filament-bound cross-bridges swing out toward thin filaments, leading to enhanced contractility (see later). The structure of MLC2_M_ may have a similar destabilizing effect on the IHM as that caused by phosphorylated MLC2_F,_ which may possibly explain the preponderance of protruding cross-bridges in thick filaments of masticatory fibres [44].

### 1.3. Kinetic Properties of Myosin Isoforms

Mechanical analyses of EOM [45,46,47] and single EOM fibres [48,49] have established that EO myosin is the fastest isoform, although definite evidence based on in vitro motility assays is still missing [50]. Of the isoforms found in limb and trunk muscles, 2B, 2X, 2A and β-slow myosins are in descending order of speed [51,52,53]. The α-cardiac myosin with MLCs, which is the same as ventricular myosin V1, is two- to three-fold faster than β-slow myosin containing the same MLCs (ventricular myosin V3) [54,55], placing the kinetic properties of α-cardiac myosin close to that of 2A myosin. This is consistent with the Vmax of rabbit skeletal muscle fibres containing α-cardiac myosin, which lies between the Vmax of fibres expressing β-slow and 2A myosin [56]. The kinetics of β-slow myosin are more sensitive to temperature than those of α-cardiac myosin [54,57] and limb fast myosins [36,48]. This implies that β-slow myosin has a higher activation energy, reflecting a more stable attached cross-bridge, an increased time that β-slow myosin is bound to actin and a slower ADP dissociation rate from the actin–myosin complex [58]. These characteristics underlie the low tension cost of β-slow skeletal fibres [59] and hearts [60] expressing the same β-slow myosin.

The velocities of shortening of newborn fast and slow rat limb muscles are virtually the same as those of adult slow muscle [61]. Since these muscles express predominantly neonatal myosin [24,62,63], the Vmax of fibres with neonatal myosin is thus close to that of β-slow myosin. However, model analysis of human neonatal myosin suggests that it shares some kinetic characteristics with fast myosins [64]. *MYH15* myosin, an orthologue of amphibian and avian ventricular MyHC, is uniquely expressed in orbital fibres of rat EOMs [21]. The kinetics of embryonic and *MYH15* myosins have not been specifically determined but are likely to be as slow or slower than those of β-slow myosin [65]. The very slow contractions of multiply-innervated EOM fibres of rabbits [66] and cats [67,68,69] can be attributed to the presence of slow-tonic myosin. This myosin is closely related to the myosin in slow-tonic fibres in birds and amphibians [70,71], which produce very slow contractures [72,73]. Recent kinetic analysis of recombinent human slow-tonic myosin revealed that its maximal actin activated ATPase activity is half that of human β-slow myosin and that a high proportion of the slow-tonic myosin heads are in the super-relaxed (SRX) state with very low ATPase activity [74].

Kinetic properties of myosin isoforms are principally determined by the structure of their heavy chains. An important determinant of kinetic properties of specific myosin isoforms is the body size of the animal, as large animals have slower myosins. The Vmax of β-slow MyHC is scaled to (body mass)^−0.175^ [75]. Analysis of the motor domain sequences of β-slow MyHCs of 67 mammals identified 16 sites associated with changes in body mass, suggesting that mutations in these sites are responsible for the changes in myosin’s kinetic properties. This was suported by the observation that substituting nine of these sites that differed between rat and human β-slow MyHC into the human β-slow MyHC to match the rat sequence enhanced the kinetics of the chimeric myosin to match that of the rat [76]. Much future work will be needed to identify structural differences responsible for controlling the kinetic properties of other myosin isoforms.

### 1.4. Modulation of Contractility by Thick Filament Proteins

While contractile properties of muscle fibres are primarily determined by the myosin isoform, molecular interactions of myosin with itself and with other components of the thick filament, titin and myosin-binding protein-C (MyBP-C) also play a role. On either side of the thick filament bare zone, most myosin heads are bound to the core of the thick filament, forming an ordered helical array; the two heads of each myosin molecule interact with each other and are folded back against the thick filament, forming a stable IHM. Titin extends from the Z-disc to the M-line in the center of the bare zone, interacting with the thick filament along the way in each half sarcomere, while MyBP-C is located in the central third of each half sarcomere called the C-zone. Myosin heads forming the IHM are in an SRX state with very low Mg-ATPase activity [77] and are in the OFF state, not being competent to interact with activated thin filaments. However, a proportion of myosin heads in resting muscle are not attached to the thick filament but are swung out toward the thin filament in a disordered relaxed (DRX) state with much higher Mg-ATPase activity relative to the SRX state. These cross-bridges are in the ON state and are competent to bind to activated actin filaments. Upon thin-filament activation by Ca^2+^, these ON cross-bridges bind to actin, producing tension and inducing stress along the thick filament. Thick filament stress destabilizes OFF cross-bridges into the ON state and causes them to interact with actin, generating more tension in a positive feedback process termed mechano-sensing [78,79,80,81]. Mechanisms that disrupt the IHM would also increase the population of ON cross-bridges and generate more stress on the thick filament, thereby modulating the contractile properties of muscle fibres through mechano-sensing. This occurs during post-tetanic potentiation in fast fibres in which MLC2_F_ is phosphorylated during a tetanus by Ca^2+^–calmodulin-activated myosin light-chain kinases [82,83]. X-ray diffraction analysis revealed that after phosphorylation of MLC2_F_, more myosin heads are shifted away from thick filaments toward thin filaments [84], thereby enhancing the post-tetanic twitch.

MyBP-C can also influence contraction by modulating the IHM. There are four fibre-type-specific MyBP-C isoforms: fast (fMyBP-C), slow (sMyBP-C), cardiac (cMyBP-C) [85] and masticatory (mMyBP-C) [86,87]. The existence of mMyBP-C has largely been ignored in the field. Fast muscle fibres express both fMyBP-C and sMyBP-C in roughly equal proportions, while slow muscle fibres express virtually only sMyBP-C [88], and mMyBP-C is expressed specifically in masticatory fibres of jaw-closing muscles. MyBP-C is an elongated, flexible polypeptide made up of a series of globular immunoglobulin (Ig)-like domains and fibronectin type 3 (Fn3) domains. sMyBP-C and fMyBP-C both contain 10 Ig/Fn3 domains, with a fibre-type-specific MyBP-C motif inserted between the first and second Ig domains in the N-terminal region. The C-terminal region of MyBP-C is tightly bound to the thick filament by interacting with myosin rods and with titin. cMyBP-C and sMyBP-C have phosphorylation sites in their N-terminals. These N-terminals can bind to either thick or thin filaments depending on their phosphorylation status. In cardiac muscle, the N-terminals of unphosphorylated cMyBP-C stabilize the OFF state of cross-bridges. Phosphorylation of cMyBP-C by cAMP-activated protein kinases, which occurs following β-adrenergic stimulation, destabilizes the myosin IHM, enhancing the number of ON cross-bridges and thus cardiac contractility [89], thereby accounting largely for the inotropic response to β-adrenergic stimulation of the heart [90].

Less is known about the roles of sMyBP-C and fMyBP-C in modulating skeletal muscle contractility. Analysis of ATPase activity using single-molecule fluorescence imaging techniques along the thick filament in β-slow skeletal muscle sarcomeres suggests that sMyBP-C does influence the distribution of SRX and DRX cross-bridges [91]. Mechanical analysis of transient force overshoot in slow muscle using a relax and restretch protocol suggests a model in which it is the unphosphorylated N-terminals of sMyBP-C which bind to thin filaments, forming links between filaments which retard filament sliding. Phosphorylation of N-terminals of sMyBP-C by cAMP-activated protein kinases detaches them from thin filaments and binds them to thick filaments. This releases the links between thick and thin filaments, thereby facilitating filament sliding. The binding of phosphorylated N-terminals to thick filaments leads to an enhanced rate and magnitude of force development presumably by turning OFF cross-bridges into the ON state [92].

Relatively little is currently known about the role of MyBP-C in the modulation of craniofacial muscle contraction. All EOM fibres have been reported to express sMyBP-C [93]. A jaw-specific mMyBP-C is expressed in masticatory fibres of carnivores [86,87], and a single unidentified MyBP-C isoform which has an electrophoretic mobility similar to that of sMyBP-C is found in human masseter muscle [94]. MyBP-C expression in laryngeal muscles has yet to be investigated.

## 2. MyHC Expression Repertoires of Craniofacial Muscles

Across mammalian species, the specific MyHC isoforms expressed in muscle fibres of a given animal are matched to the functional requirements of the structure that the muscle fibres move. The repertoire of limb and truck muscles comprises β-slow, 2A, 2X and 2B MyHCs expressed in β-slow, 2a, 2x and 2b fibres, respectively. Craniofacial muscles have very different repertoires for MyHC gene expression compared to muscles of somitic origin. Adult EOMs express nmMyH IIB MyHC and all sarcomeric muscle MyHC isoforms except the masticatory MyHC [95]. This means that EOMs express 11 kinetically distinct MyHC isoforms. The extremely wide range of kinetic properties of these MyHC isoforms together with their complex distribution in various fibre types and variations along fibre length enable EOMs to execute very fast saccades without overshoot, ripple-free eye fixation and various other types of eye movements.

MyHC isoforms expressed in jaw-closing muscles of different mammalian taxa vary widely to match the masticatory requirements of the wide range of diets and feeding habits of these animals [3]. These MyHCs include the high-force and highly jaw-specific masticatory MyHC in carnivores and other animals, α-cardiac and β-slow MyHCs in herbivores and limb MyHCs in omnivores.

Laryngeal muscles express limb muscle MyHC isoforms, with the profile of MyHCs shifting toward faster isoforms in smaller animals; the smallest animals also express the superfast EO MyHC. This pattern of MyHC expression enables the laryngeal muscles of animals of varying body mass to cope with respiratory rate changes associated with increasing basal metabolic rate related to decreasing body mass [96].

The MyHC expression repertoire of craniofacial muscles is related to their allotype. Table 2 lists the MyHC expression repertoires of extraocular, masticatory and laryngeal muscle allotypes.

## 3. Craniofacial Muscle Allotypes

The capacity of craniofacial muscles to express different isoforms of MyHC and other myofibrillar proteins is intrinsic to the lineage of myoblasts which give rise to them. Cat temporalis muscle regenerated from temporalis muscle satellite cells and innervated by limb fast-muscle nerve expresses masticatory MyHC [97] and the jaw-specific mMyBP-C and tropomyosin (m-Tm) [98] rather than limb muscle isoforms of these proteins. Even uninnervated regenerated temporalis muscle expresses masticatory MyHC [99], and myotubes derived from tissue cultures of satellite cells from cat temporalis muscle express masticatory MyHC, mMyBP-C and m-Tm [87]. Further, transplanted cat temporalis muscle innervated by limb slow-muscle nerve expresses not limb β-slow myosin but an immunochemically distinct jaw-specific β-slow myosin [97,98], with β-slow MyHCs associated with masticatory MLC1_M_ and MLC2_M_ [36,100]. These results indicate that the capacity to express masticatory myosin, mMyBP-C, m-Tm and jaw-specific slow myosin is intrinsic to the temporalis myoblast lineage. The term allotype has been used to classify muscles from different body regions with different repertoires for expressing myosins and other myofibrillar proteins.

The status of EOM as a distinct allotype has also been examined. Initial experiments on myotubes in cultured mouse EOM satellite cells and on mouse limb fast muscle with incorporated EOM satellite cells showed that they failed to express EO and slow-tonic MyHCs specific to EOMs [4]. Regenerated rabbit EOM innervated by limb fast-muscle nerves also do not express EO MyHC. However, regenerated EOM innervated by the oculomotor nerve does express EO MyHC [95], confirming that myogenic cells of EOM can express the EOM-specific MyHC, and that EOM is also a distinct allotype. The failure of regenerated EOM innervated by limb muscle nerves to express EO MyHC may be due to the absence of a neural impulse pattern necessary for its expression. The oculomotor nerve can deliver very high-frequency impulses [101], especially during saccades [102,103]. High-frequency impulses may be necessary for inducing EO MyHC expression but are absent from limb muscle nerves. It remains for future research to confirm that very high-frequency impulses per se are specifically required for inducing EO MyHC expression in regenerated EOMs. Consistent with EOM being a distinct allotype, these muscles show different susceptibilities to muscle disease compared to limb muscles, being spared in Duchenne muscular dystrophy [104] but are early and prominent targets in myasthenia gravis [105].

Several observations suggest that laryngeal muscles are also allotypically distinct from other craniofacial and limb muscles. The capacity of rat thyroarytenoid muscle to express EO MyHC but not β-slow and 2A MyHCs is unaffected by cross-innervation by the nerve to the sternohyoid, a somitic muscle which expresses limb MyHC but not EO MyHC [106]. Although the cricothyroid (CT) in most mammals expresses limb muscle myosin isoforms [107], it is highly specialized in bats for producing ultrasonic sound pulses for echolocation. Bat CT is an extremely fast muscle [108] in which the sarcoplasmic reticulum volume occupyies 30% of the fibre volume [109] and the mitochondrial density is enhanced [110]. The CT is also differentially susceptible to disease compared with other laryngeal muscles. In a mouse model of Duchenne muscular dystrophy, the CT is affected while the other laryngeal muscles are spared [111,112,113]. Videolaryngoscopic examination of laryngeal muscle function in patients with Duchenne muscular dystrophy suggests that vocal fold adductors and abductors are also spared in this disease [114]. These observations show that the CT and other laryngeal muscles differ in allotype.

This review will examine the developmental origins of the major craniofacial muscles, the expression, regulation and functional significance of the MyHCs that they express and changes in MyHC expression during phylogeny. It will focus on how differences in MyHC expression meet the diverse functional demands of the various organs that they move and how changes in their functional demands during phylogeny are met by changes in MyHC expression.

## 4. Extraocular Muscles

### 4.1. Developmental Origins of EOMs

Mammalian EOMs develop from two components of the head mesoderm. The first component is of prechordal mesodermal origin and gives rise to four EOMs (medial rectus, inferior rectus, superior rectus and inferior oblique) innervated by the oculomotor nerve. The other portion arises from the paraxial mesoderm related to the first branchial arch, giving rise to the superior oblique muscles, innervated by the trochlear nerve, and the lateral rectus, innervated by the abducens nerve [5,7,115] (see Table 1). Normal development of the EOMs requires input signals from both the developing eye and surrounding cranial neural crest cells [7]. Expression of the Pitx2 gene is necessary for the specification of EOM myoblasts [116], in contrast to the corresponding requirement of Pax3 for somitic myoblasts [117]. Pitx2 activates myf5, myf6 and MyoD1 to commit premyogenic cells to EOM myogenesis [10,118], while myogenin initiates EOM myogenesis. Postnatal knockout of the Pitx2 gene shows that the continued expression of Pitx2 in adult EOM fibres is necessary for the maintenance of EOM-specific characteristics such as polyneuronal innervation and slow-tonic MyHC expression [119,120].

### 4.2. Ontogeny of EOMs

Changes in MyHC expression in EOMs during perinatal development in rats [121,122] and following postnatal experimental manipulations [123,124] have been reported. Early on during EOM myogenesis, primary and secondary myotubes emerge in succession from primary and secondary myoblasts, just as in developing muscles elsewhere. At E16 (embryonic day 16), only primary myotubes are seen, expressing β-slow MyHC strongly and embryonic/neonatal MyHCs weakly. Secondary myotubes appear at E17, expressing embryonic/neonatal MyHCs but not β-slow MyHC. Thus far, the pattern of MyHC expression in these myotubes resembles that in limb muscles. Subsequently, the division of EOMs into orbital and global layers begins to emerge. Slow-tonic myosin first appears only in slow primary myotubes at E20 in Sprague-Dawley rats [121] but has been reported to be expressed in all fibres of fetal EOM in these rats; the expression of this myosin disappears in most fibres, except the slow-tonic fibres during early postnatal life [21]. In rats at birth, β-slow MyHC is exclusively localized to primary fibres and embryonic/neonatal MyHCs are present in all fibres, while fast MyHCs are expressed only in secondary fibres [122]. In postnatal mice, β-slow MyHC is exclusively localized to evenly distributed small fibres presumably of primary myotube origin, while fast MyHCs are expressed in larger surrounding fibres presumably of secondary myotube origin [125]. In adult rabbit EOM, β-slow MyHC is also exclusively localized to multiply-innervated fibres (MIFs) and fast MyHCs are exclusively localized in singly-innervated fibres (SIFs). These results suggest that primary myotubes become MIFs, while secondary myotubes become SIFs.

During the first 2–3 weeks of postnatal life in rats, dramatic changes in MyHC expression occur: neonatal and embryonic MyHCs are downregulated in the global fibres but are retained in the orbital fibres [122], and 2A/2X [121], EO [126], nmMyH IIB [127] and MYH15 [21] MyHCs progressively emerge. These changes are associated with progressive maturation of oculomotor neurons and the visual nervous system, and eye opening at around 2 weeks postnatally. During this period, the ratio of tonic to phasic discharge patterns of motoneurons increases about ten-fold and the maximal frequency of discharge of oculomotor neurons rises from 150 to 420 Hz [101]. Postnatal dark rearing decreases the proportion of fibres expressing β-slow and fast MyHCs and reduces the expression of EO MyHC mRNA [123], while chemical destruction of vestibular hair cells postnatally increases the proportion of developmental and fast MyHCs and decreases EO MyHC mRNA expression [124]. These results strongly suggest that postnatal changes in the oculomotor impulse pattern via inputs from visual and vestibular pathways induce the expression of various MyHCs that emerge postnatally.

In mice with postnatal knockout of Pitx2, the expression of β-slow and slow-tonic MyHCs, along with troponin I and troponin T, is reduced during the first 3 weeks after birth. This is associated with the loss of *en grappe* neuromuscular junctions and polyneuronal innervation, suggesting that the expression of β-slow and slow-tonic MyHCs is regulated by the tonic motoneurons normally innervating MIFs [120].

Our current understanding of EOM development is woefully incomplete. For example, we do not know the cellular and molecular basis of the division of EOMs into global and orbital layers, nor the basis for the variation in MyHC expression along the fibre length. In limb muscles, the postnatal surge of thyroid hormone [128] and the emergence of different neural impulse patterns play key roles in muscle fibre maturation [129]. While the role of the nerve is proposed below, there has apparently been no study of the possible role of thyroid hormone in postnatal MyHC expression in EOMs.

The integrity and synaptic connectivity of different types of ocular motoneurons are sustained by different retrogradely transported muscle-derived neurotrophins [130] which specify their synaptic inputs and thus the efferent neural impulse patterns (see later). Oculomotor neurons depend on these trophic factors for their survival during development [131]. Neonatal EOMs richly express these neurotrophic factors, and exogenous replacement of these factors has been shown to significantly rescue extraocular motoneurons from axotomy-induced cell death [132]. Myogenic cells isolated from EOMs also express high levels of neurotrophins [133]. It is hypothesized here that synaptic inputs of slow-tonic and phasic motoneurons innervating MIFs and SIFs, respectively, are modulated during development by different target-derived neurotrophins from contact with primary and secondary myotubes, respectively. In other words, through the different neurotrophic factors retrogradely transported to their motoneurons, primary and secondary myotubes may choreograph the tonic and phasic types of the neuronal impulse pattern that they, respectively, receive. These patterns would, according to the impulse pattern hypothesis (see below), control MyHC expression in a frequency-dependent manner during postnatal development as well as the maintenance of the MyHC expression pattern during adult life.

### 4.3. Functional Demands on EOMs

The functional demands on EOMs are highly complex but are very similar across species, in sharp contrast to widely varied demands on jaw-closing muscles. The nervous system has evolved mechanisms to control the EOMs to move the two eyes as a functional unit to optimize vision [134,135]. To obtain good vision, both eyes must fixate on the same object of interest. If the head moves, the eyes must make precise compensatory movements in the opposite direction to maintain visual acuity. The vestibulo-ocular system and the optokinetic system must compensate for large and small head motion by producing perfectly matched counter rotations of the eyes. There are two gaze-shifting mechanisms: (1) the saccadic system, which rapidly shifts gaze from one point to another, and (2) the smooth pursuit system, which allows the eyes to track a moving target. To keeps both eyes aligned with visual targets at various distances from the eyes, there is the vergence system, which is better developed in front-eyed animals and predators than lateral-eyed animals and preys. These neural systems converge on ocular motoneurons which control the six EOMs [136], each of which contain six different fibre types [137] to cater for the functional demands on EOMs. Functional demands are greatest during a saccade, in which the eyes rapidly and accurately move from one fixation point to another in an almost linear fashion without overshoot. Eye movements must also satisfy Listing’s law which governs the axis of rotation of the eye in three-dimensional space [138].

### 4.4. EOM Fibre Types

To meet the very complex functional demands, EOM fibres are highly specialized and differ substantially from other craniofacial and limb muscle fibres in many characteristics, including innervation, fibre types, MyHC composition and gene expression profile [137]. They have MIFs resembling slow-tonic fibres in amphibians and express unique MyHCs [21,139]. Single EOM fibres co-express multiple MyHC isoforms which may vary along the fibre length [140,141,142,143,144]. The pattern of gene expression in EOMs differs from that of limb muscles and has been documented by proteomics [145] and mRNA profiling [146,147,148].

The fibres of EOMs are organized into two layers: a thin orbital layer facing the orbit and a thicker global layer facing the globe [149,150]. Both layers are made up of (1) SIFs with large *en plaque* nerve endings and (2) MIFs with small *en grappe* endings. Using ultrastructural and histochemical criteria, fibres in the orbital (o) and global (g) layers can be classified into six fibre types: oSIF and oMIF in the orbital layer, and gSIF-red, gSIF-intermediate, gSIF-white and gMIF in the global layer [137].

The oSIFs have small myofibrils, well-developed T-tubules and a moderately developed sarcoplasmic reticulum, rich in mitochondria and oxidative enzymes, suggesting that they are fatigue-resistant. The three types of gSIFs vary in fibre diameter; SIF-red fibres are small and rich in mitochondria and oxidative enzymes, suggesting high fatigue resistance, gSIF-white fibres are large, poor in mitochondria but rich in glycolytic enzymes, suggesting high fatigability, while oSIF-intermediate fibres have intermediate values for these characteristics [137].

MIFs generally have large myofibrils, few mitochondria, poorly developed T-tubules and a sarcoplasmic reticulum. While oMIFs receive an extra *en plaque* endplate in the middle of the fibre and can generate non-overshooting action potential in this local region [151] and thus can produce twitch responses [151,152], gMIFs cannot initiate an action potential, resembling the non-twitch, slow-tonic fibres found in amphibian limb muscles [72,73].

The gMIFs in some animals have innervated structures called palisade endings or innervated myotendinous cylinders [153] at their myotendinous junctions. Palisade endings resemble immature Golgi tendon organs and are found in all front-eyed mammals studied but are absent in some lateral-eyed mammals [154]. They are innervated by axon extensions of nerve fibres of oculomotor neurons innervating gMIFs [155,156]. The function of palisade endings is highly controversial, with many researchers regarding them as proprioceptive receptors [157], but others believe that they are motor in nature [158,159]. It has been hypothesized that palisade endings in front-eyed animals are mechano-receptors that can be activated by force generated by gMIF contraction as well as by passive stretch to generate axon reflexes that activate gMIF motor units to enhance their force output during fixation, saccades and vergence [95]. Since gMIFs are multiply-innervated fibres, the gMIF/palisade ending complex acts like a motoneuron with afferent synaptic inputs from several gMIF motoneurons, integrating not their endplate potentials, but summing their tension contributions to stimulate the palisade ending to generate neural impulses to produce tension from the gMIF motor unit of which it is a part.

The patterns of gene expression in orbital and global layers differ [160,161]. Fibres of the orbital layer are short and insert onto connective tissue pulleys attached to the orbital wall [162], though some orbital fibres do insert onto the eyeball [144,163] and some global fibres in human and monkey EOMs also insert onto their pulleys [164]. Orbital fibres have long been implicated in fixation [165,166,167], but many researchers currently hold that orbital fibres move the pulleys which modulate the direction of rotation of the eyeball to implement Listing’s law [168,169] and that only global fibres move the eyeball. Little is currently known about the cellular and molecular bases for the division of EOMs into orbital and global layers.

### 4.5. Ocular Motoneurons Innervating MIFs and SIFs

Motoneurons supplying SIFs and MIFs are located in different regions of the ocular motor nuclei. Motoneurons innervating SIFs are located within the classical ocular motor nuclei, whereas those innervating MIFs are located in the periphery of these nuclei [170]. Two populations of neurons have been identified in this peripheral region in the monkey oculomotor nucleus [171]; a population of small, multipolar neurons innervate palisade endings and their gMIFs [172,173], while the other population of round or spindle-shaped cells probably innervate oMIFs.

The firing pattern of ocular motoneurons spans a wide spectrum ranging from tonic to phasic [174]. SIF motoneurons have high thresholds and high velocity sensitivities, are deployed infrequently and deliver a range of high-frequency bursts during saccades. MIF motoneurons participate in all types of eye movements as SIF motoneurons but are recruited earlier and show lower eye position and eye velocity sensitivities than SIF motoneurons, and fire at relatively low frequencies [103].

Neurotrophins play important roles during neuronal development and in adult life, when they act as mediators of synaptic and morphological plasticity [175,176]. The integrity and synaptic connectivity of ocular motoneurons are sustained by different muscle-derived neurotrophins, as in the case of spinal and other craniofacial motoneurons [130]. The burst and step features of ocular motoneuron impulses are generated by different synaptic inputs and are maintained by different neurotrophins retrogradely transported from the muscle fibres that they innervate [135]. Muscle fibres [177], including EOM fibres [132], abundantly express brain-derived neurotrophic factor (BDNF), neurotrophin-3 (NT-3) and nerve growth factor (NGF). Following axotomy, ocular motoneurons lose most of their synapses, resulting in reduced impulse frequency and reduced position and velocity sensitivities [175]. The administration of neurotrophins to the proximal nerve stump restores the afferent synaptic contacts and firing characteristics of axotomized ocular motoneurons, BDNF restores tonic firing and NT-3 restores saccadic bursts [175], while NGF has an influence on both types of activity [178]. Since myogenic cells isolated from EOMs express high levels of neurotrophins [133], these interactions between muscle and nerve would presumably occur following the initial innervation of primary and secondary myotubes by MIF and SIF motoneurons, respectively, resulting in their different neural impulse patterns that regulate the postnatal changes in MyHC expression in MIFs and SIFs.

### 4.6. Longitudinal Variations in EOM Fibre Characteristics and Their Functional Significance

An outstanding feature of the oSIFs and oMIFs is that unlike gSIFs, they show marked systematic variations along their lengths in structural and functional features, including fibre diameter, [163,179,180] myofibrillar size [163,180], mitochondrial volume [180,181], sarcoplasmic reticulum volume [180], myosin ATPase histochemistry [182] and expression of MyHC genes [140,141,142,143,144]. The pattern of MyHC expression of these fibres is listed in Table 3. The differences in MyHC composition of oSIFs and oMIFs indicate that they differ in speed, but these fibre types share the common feature that they express their respective fastest MyHC(s) in their central segments, flanked by end segments co-expressing several slower MyHCs.

It has recently been proposed that the functional significance of variations in MyHC expression of the central and end segments of oSIFs and oMIFs is that they are instrumental in linearizing the saccade [144]. At the commencement of a saccade, the antagonist must relax and be lengthened rapidly to allow the agonist to accelerate the eyeball. The relaxation rates of the central segments of the antagonist oSIF and oMIF units containing EO and α-cardiac MyHCs, respectively, would rapidly yet smoothly reduce the viscous load on the contracting agonist motor units, thus facilitating the initial angular acceleration of the eyeball. The twitch-like feature of the endplate region of oMIFs probably also contributes to faster relaxation. The relaxation rate of canine oMIFs is enhanced by the association of MLC1_E/A_ with β-slow MyHC in oMIFs, whereas in the gMIFs, β-slow MyHC is associated with MLC1_S_ [183]. Without the fast, central segments, the slower cross-bridges of the antagonist fibres would resist rapid lengthening by developing high tensions [184]. Deceleration of the eyeball toward the end of the saccade is promoted by the end segments of antagonist oSIFs and oMIFs, which express a range of kinetically slow MyHCs. Stretching these slow, attached cross-bridges in the end segments of oSIF and oMIF would produce a high but progressively decreasing viscous load to decelerate the eyeball.

Variation in MyHC expression along the length also occurs in gMIFs. These fibres express α-cardiac and β-slow MyHCs in their central segment but the end segments additionally express slow to very slow isoforms: embryonic, slow-tonic and nmMyH IIB MyHCs. At the end of a saccade, passive tension of the stretched antagonist is transmitted along gMIFs to the palisade endings due to the very slow relaxation rates of these MyHCs. The resulting axon reflex helps to augment the slowly rising postsaccadic tension of the antagonist to match the postsaccadic fall in tension of the agonist to maintain fixation at the new location [95].

### 4.7. Regulation of MyHC Expression in EOM

The role of neural impulse activity on MyHC regulation in the EOM has been studied using botulinum toxin-mediated denervation [185] and in transplantation studies of EOMs regenerating under the influence of different types of nerves [95]. Botulinum toxin blocks acetylcholine release at the neuromuscular junction, bringing about a prolonged effective denervation followed by a slow recovery of function associated with the sprouting of terminal axons and the formation of new neuromuscular junctions. Following administration of this toxin to EOM in adult rats, SDS-PAGE revealed that EO MyHC was permanently abolished and 2B MyHC drastically reduced, while 2A and 2X MyHCs dramatically increased 5 months after treatment with the toxin, with only partial recovery of the proportion of 2B MyHC after 8 months [185]. The complete loss of EO MyHC and the very slow partial recovery of the MyHC expression profile may be related to the loss of high-frequency impulses. This may be due to the failure of the regenerating endplates to transmit high-frequency impulses but may also be related to a prolonged impairment of the retrograde transport of neurotrophins to sustain the appropriate neural inputs to the oculomotor neurons to generate the normal range of impulse patterns.

Rabbit EOMs regenerating in situ and in fast limb muscle beds after freeze injury were compared using immunohistochemical analyses [95]. Regenerated EOMs innervated by limb fast-muscle nerves strongly express the relatively slow neonatal, β-slow/slow-tonic and 2A MyHCs but not the faster MyHCs, even in the long term, whereas in situ regenerated EOMS innervated by the oculomotor nerve are able to express not only the slower MyHCs within the repertoire, but also the much faster 2X, 2B and EO MyHCs. These experiments suggest that the capacity of in situ regenerated EOM to express the fast MyHCs is associated with the availability of a range of high to very high frequency impulses from oculomotor neurons.

The expression of the full range of MyHCs in in situ regenerated EOM may have been facilitated by the selective reinnervation of regenerating SIFs and MIFs by their respective motoneurons. Selective reinnervation of slow-tonic and fast-twitch muscle fibres by their respective original nerve fibres has been reported in avian [186] and amphibian [187] muscles during reinnervation after their denervation. It is thus possible that tonic MIF motoneurons would selectively reinnervate regenerating MIFs while phasic SIF motoneurons selectively reinnervate regenerating SIFs. This will result in the retrograde transport of appropriate neurotrophic factors from MIFs and SIFs to ensure the delivery of the normal wide range of neural impulse frequencies and patterns of ocular motoneurons [174] to induce the expression of the full range of EOM MyHCs in the various fibre types. Low-frequency tonic impulse patterns are postulated to regulate the expression of MyHCs at the slower end of the kinetic range (embryonic, neonatal, slow B, β-slow and 2A MyHCs) and high-frequency phasic impulse patterns regulate the expression of the fastest MyHCs within the EOM repertoire, namely, 2X, 2B and EO. Motoneurons with firing patterns between these extremes show phasic and tonic activities to varying degrees and are postulated to control the expression of MyHCs in a frequency-dependent manner; the more phasic impulse patterns with higher frequencies control the faster MyHCs, while the more tonic ones with lower frequencies regulate the slower MyHCs [95]. During spontaneous eye movements, the discharge characteristics of oculomotor neurons are characterized by a high-frequency pulse followed by a slide of decreasing frequency and then a lower-frequency step [174]. The delivery of impulses with a variety of frequencies to the same motor unit could explain the co-expression of multiple MyHCs in gSIFs.

The functional significance of the above impulse pattern hypothesis is that the very signals used to drive the various ocular movements are instrumental in equipping the different EOM motor units with appropriate MyHCs and thus the contractile properties needed to accomplish their tasks. In EOM, the low-frequency impulse patterns of tonic motoneurons required for generating the mechanical characteristics for controlling eye fixation, tracking and vergence would induce the expression of MyHCs within the slow end of the kinetic spectrum, leading to relatively slow-contracting, low-power and economic motor units appropriate for these tasks. The high-frequency bursts of phasic motoneurons necessary for generating the mechanical features of large saccades and quick phases of vestibulo-ocular reflexes would induce the fastest MyHCs, leading to fast and powerful motor units suitable for rapid eye movements.

The notion that the impulse pattern needed to generate the desired mechanical activity is instrumental in inducing the appropriate MyHC also seems to apply in the case of limb muscle. The neural impulse pattern sent to the slow soleus muscle fibres in rats needed to generate sustained muscle tension is characterized by a sustained activity at low frequencies [188]. This chronic low-frequency stimulation (CLFS) controls β-slow MyHC expression in the soleus muscle and can convert cat masticatory fibres in temporalis muscle and fast fibres in laryngeal muscle into β-slow fibres (see later). Impulses delivered to the fast extensor digitorum longus (EDL) muscle, which predominantly expresses the three fast MyHCs, are characterized by high frequencies and have two subtypes. One involves short bursts of high-frequency impulses needed to generate powerful, brief bursts of tension; this impulse pattern supports 2B MyHC expression in denervated EDL muscle. The other subtype is characterized by long bursts of high-frequency impulses needed to generate prolonged fast contractions; this pattern supports the expression of 2A/2X MyHCs but suppresses 2B MyHC in denervated EDL muscle [189].

The above discussion does not address the localizations of slow MyHCs at the end segments and of fast MyHCs in the central segments of oMIFs, oSIFs and gMIFs, which are functionally very important. Future work needs to investigate how MyHC expression in the central and end segments of these fibre types is differentially regulated.

Relatively little is known about the role of thyroid hormone in MyHC expression in EOMs. It seems likely that thyroid hormone would facilitate the postnatal maturation of EOMs as it does in limb and other craniofacial muscles. In adult rabbits, hyperthyroidism significantly decreases EOM mass, the total number of fibres and the mean cross-sectional area of the fibres. Immunohistochemical analysis using antibodies against fast, slow, embryonic and neonatal MyHCs revealed that the percentages of neonatal and embryonic MyHC-positive fibres decreased, while the percentages of β-slow-positive fibres significantly increased in both global and orbital layers and fast MyHC-positive fibres were significantly reduced in both the central and end segments of the orbital layer fibres [190]. Since EOM fibres express 11 MyHCs and fibres co-express multiple MyHCs and the antibodies used in this study were deficient in number and specificity, these results are difficult to interpret. It is not possible to assess whether thyroid hormone shifted the expression of 2A MyHC toward 2B MyHC as it does in limb, jaw and laryngeal muscles (see later). The onerous task of clarifying the role of thyroid hormone in MyHC expression in EOMs would require the use of a comprehensive set of antibodies specific against each of the 11 MyHC isoforms and would have to include an analysis of changes along the length of various fibre types!

## 5. Jaw Muscles

### 5.1. Developmental Origins of Jaw Muscles

Muscles that move the jaw develop from myogenic cells from the first branchial arch. These muscles fall into two functional groups: jaw-closers (temporalis, masseter, medial pterygoid and the superior head of the lateral pterygoid) and jaw-openers (mylohyoid, anterior digastric and the inferior head of the lateral pterygoid). Both groups of muscles are innervated by the mandibular branch of the trigeminal nerve (cranial nerve V). Jaw closers develop from cells in the dorsal portion of the first arch which derive from the cranial paraxial mesoderm, while jaw-openers develop from mesodermal cells in the ventral portion of the first arch which derive from the lateral splanchnic mesoderm [191] (see Table 1). Pitx2 regulates transcription factors that specify first-branchial-arch myoblasts. It acts to assure the expression of premyogenic genes Tbx1, capsulin (Tcf21) and MyoR (musculin, Msc) in the first-arch-derived muscles [192]. The Tbx1 gene commits cranial paraxial mesoderm progenitor myoblasts to pharyngeal muscles [4,193], while the expression of MyoR or of capsulin is necessary for the myogenesis of the jaw-closer muscles. In the absence of both of these genes, jaw-closer muscles are absent, but the jaw-opener muscles develop normally [194]. Besides giving rise to jaw-opening muscles, cells of the lateral splanchnic mesoderm also have cardiogenic potential and are characterized by the strong expression of transcription factor islet1 [195]. A set of transcription factors expressed in lateral splanchnic mesoderm progenitors form a regulatory gene network that coordinates normal craniofacial muscle and cardiac development [196].

### 5.2. Functional Demands on Mammalian Jaw Muscles

Jaw-closer and jaw-opener muscles represent two distinct muscle allotypes with different myosin expression profiles. Jaw-closers provide the power behind the specialized bony and dental apparatus designed for the procurement and mastication of food. Across mammalian species, in sharp contrast to EOMs, functional demands on jaw-closing muscles are extremely varied, having to cope with the different lifestyles, diets and eating habits of the animals. These demands range from requiring a high-force bite for hunting in carnivores to the economy of energy expenditure for sustained mastication in ruminants. Consequently, myosin isoforms expressed in jaw-closers of different species vary widely, ranging from the high-force jaw-specific masticatory myosin to one or more myosins of various speeds and powers found in limb and cardiac muscles. The functional load on jaw-opening muscles, however, is relatively constant and light, and they express myosin isoforms similar to those in limb muscles [197,198,199].

### 5.3. Fibre Types in Jaw-Closing Muscles of Carnivores

Rowlerson and co-workers first showed that muscle fibres in cat jaw-closers express a jaw-specific masticatory myosin with unique MyHCs and MLCs. The genes for cat masticatory MyHC (*MYH16*) [23] and MLC2_M_ (*MYL5*) [43,200] have been sequenced and shown to be of very ancient origin. The masticatory MyHC gene is the first among mammalian MyHC genes to diverge from ancestral forms [23]. Masticatory MLC1_M_ (encoded by *MYL4*) is identical to the embryonic/atrial MLC1 (MLC1_E/A_) [33]. Cat masticatory fibres also express a jaw-specific mMyBP-C [86,87] and m-Tm [87,201].

Jaw-closers of carnivores also contain jaw-specific β-slow fibres which differ histochemically [197] and immunohistochemically [202] from limb β-slow fibres. These jaw slow fibres express the same β-slow MyHC as in limb slow fibres, but differ in MLC composition, expressing MLCs identical to those associated with masticatory MyHC [36,100].

As will be discussed below, masticatory fibres are associated with a very high force of contraction to generate a strong bite, while jaw slow fibres are able to relax rapidly to permit the jaw to open wide rapidly prior to jaw closure.

### 5.4. Ontogeny of Jaw-Closing Muscles in Carnivores

In dogs, the temporalis muscle prenatally expresses developmental MyHCs and MLCs similar to those of limb muscles during early myogenesis but begins to express masticatory MyHC and MLCs two weeks postpartum [203]. Detailed immunohistochemical analysis of cat masseter muscle during perinatal development shows that there are four developmentally distinct myotube types, each characterized by a specific sequence of myosin gene expression during perinatal development [204]: (1) slow primaries which initially express embryonic and β-slow MyHCs but not neonatal MyHC and end up expressing β-slow MyHC at maturity, (2) masticatory primaries that initially express embryonic, neonatal and β-slow MyHCs but postnatally replace these with masticatory MyHC, (3) slow secondaries, which initially also express embryonic and neonatal MyHCs but postnatally replace these with β-slow MyHC with or without masticatory MyHC, ending up expressing only β-slow MyHC at maturity, and (4) masticatory secondaries, which initially express embryonic and neonatal MyHCs, which are postnatally replaced by masticatory MyHC [204] (see Table 4). In the developing cat posterior temporalis, which is composed homogeneously of masticatory fibres in the adult, only masticatory primary and masticatory secondary myotubes are present [205]. The posterior temporalis is homologous to the white region of limb fast muscle which is composed homogeneously of fast fibres and develops from fast primary and fast secondary myotubes [206].

The four types of myotubes in developing cat jaw-closing muscles are homologous to the four types of myotubes in developing limb muscle [206]. During limb muscle development, primary myotubes in the superficial region of fast muscles (the fast primaries) express β-slow MyHC early postnatally [207], but this is replaced by fast MyHC at maturity [207,208], presumably under the postnatal surge of thyroid hormone. The secondary myotubes in this region (the fast secondaries) begin to express 2B, 2X or 2A MyHC or a combination of these MyHCs early postnatally and continue to do so in the adult animal [209]. However, during hypothyroidism in adult cats, fast fibres of fast primary origin revert to expressing β-slow MyHC, while fast fibres of fast secondary origin express 2A MyHC [208]. The 2B MyHC-expressing fibres in the mature animal derived from fast primary and fast secondary myotubes are not the same but constitute distinct ontotypes which retain their intrinsic myotube-derived properties in adult life: under hypothyroidism, only fibres derived from fast primaries express β-slow MyHC [129].

It is likely that thyroid hormone and neural impulse patterns play the same roles in the maturation of carnivore jaw-closers as in limb muscles, and that the notion of myotube ontotype also applies to craniofacial muscles. Of the two types of myotubes in the homogeneously masticatory fibres of cat posterior temporalis, only masticatory primaries express β-slow MyHC during development [205]. I postulate that the postnatal surge of thyroid hormone is required to induce the expression of masticatory MyHC in both myotube types, and that following hypothyroidism in the adult, only fibres of masticatory primary myotube origin will express β-slow MyHC. This would be analogous to the 2b fibres of fast primary ontotype expressing β-slow MyHC under hypothyroidism [208]. The apparent muscle and fibre type specificities of the changes in MyHC gene expression in response to changes in the thyroid hormone level [210,211] are likely due to differences in muscle fibre ontotype.

### 5.5. Mechanical Properties of Masticatory Fibres of Carnivores

Since cat masticatory fibres have a very short contraction time [97,212] and masticatory myosin has a high ATPase activity [31], which normally connotes high speed [13], these fibres were thought to be superfast contracting fibres, and masticatory myosin was initially known as superfast myosin. However, the measurement of both dynamic stiffness [36] and unloaded shortening velocity [213] indicated that masticatory fibres of cats and dogs are comparable in terms of speed of contraction to limb 2a fibres.

The rapid development of force indicated by the very short contraction time may be attributed to the high Ca^2+^ sensitivity [214] and the high proportion of ON cross-bridges along thick filaments, revealed by X-ray diffraction studies [44]. A similar acceleration of the rate of force development associated with cross-bridge detachment from the thick filament occurs during post-tetanic potentiation of the limb fast-muscle twitch [215]. This is also associated with enhanced ON cross-bridges [84] due to the phosphorylation of MLC2_F,_ leading to the destabilization of the IHM on the thick filament. The swinging out of cross-bridges toward thin filaments in resting masticatory fibres may result from a structural feature of MLC2_M_ resembling phosphorylated MLC2_F_ by destabilizing the IHM. The mMyBP-C may also play a role in destabilizing the IHM and enhance Ca^2+^ sensitivity. These possibilities need to be resolved by future experiments.

Masticatory fibres develop about 40–60% more force per cross-sectional area compared with limb fast fibres [213,216]. This attribute is highly appropriate for a carnivorous lifestyle but is associated with a higher tension cost [216]. The basis of the high force is thought to be an increase in the cross-bridge attachment rate coupled with a matching decrease in detachment rate [213], but a model of masticatory myosin cross-bridge cycling kinetics that also explains the high ATPase activity of masticatory myosin and the high tension cost is lacking. The possible role of mMyBP-C in modulating cross-bridge function requires clarification by future research.

Another highly desirable property of a jaw-closing muscle for a carnivore would be the ability to develop a very high tension when stretched while active, as it would help to prevent a struggling prey from escaping. Stretching an isometrically contracting muscle leads to an increase in force [217], which has been attributed to the recruitment of myosin heads that exhibit fast attachment to and detachment from actin in a cycle that does not involve ATP splitting [218]. It is thus important for future work to determine whether the lengthening of active masticatory fibres produces higher forces than limb fast fibres.

### 5.6. Mechanical Properties of Jaw Slow Fibres of Carnivores

The fact that jaw slow myosin is associated with masticatory MLCs leads to very different kinetic properties compared with limb slow fibres. Masticatory MLC1 or MLC1_E/A_ has a weaker interaction with actin compared with slow MLC1_s_ of limb slow fibres [35]. This raises the dynamic stiffness parameter, f_min_, of jaw slow fibres to ten-fold higher than that of limb slow fibres [36]. This parameter is sensitive to rate constants for the power-stroke and the rate of cross-bridge detachment [219]. Further, the temperature sensitivity of f_min_ is reduced relative to that of limb slow fibres, indicating a reduced activation energy for the cross-bridge cycling, which reflects a reduced stability of the attached cross-bridge [36] and an accelerated cross-bridge detachment rate, leading to a more rapid rate of relaxation.

The function of jaw slow fibres in jaw-closing muscles is to produce a sustained force for holding the jaw against gravity. The functional significance of the expression of MLC1_E/A_ in jaw slow fibres in jaw-closers appears to be to enable a carnivore to rapidly open the jaw prior to seizing a prey.

### 5.7. Neural Regulation of Masticatory and Jaw Slow Fibres of Carnivores

The roles of cell lineage and innervation in regulating the expression of masticatory and jaw slow myosins and other myofibrillar proteins gleaned from transplantation experiments have been referred to earlier in connection with the notion of muscle allotype. Those experiments [97] suggest that the expression of masticatory and jaw slow myosin is regulated by the nerve connected to fast and slow muscle, respectively. The induction of β-slow MyHC and the suppression of masticatory MyHC in regenerated cat temporalis muscle innervated by the limb slow nerve [97,98] suggest that these changes are brought about by CLFS mediated by the tonic motoneurons innervating slow limb muscle. This has been confirmed by the transformation of masticatory fibres into jaw slow fibres through CLFS of the cat temporalis muscle [220]. It is likely that the molecular mechanism involved here is the same as that which transforms limb fast fibres into β-slow fibres. In this mechanism, the elevated intracellular Ca^2+^ concentration due to CLFS activates the phosphatase calcineurin, which dephosphorylates the nuclear factor of activated T cells (NFATc1) to enable it to enter the nucleus to interact with myocyte enhancer factor 2 (MEF-2C) to induce β-slow MyHC expression [221,222,223,224]. The hypothesis that jaw slow fibres are also regulated by the calcineurin/NFAT pathway requires confirmation. How CLFS suppresses masticatory myofibrillar proteins requires future elucidation.

The nerve connected to fast limb muscle may also regulate masticatory MyHC expression using the same transcription factors employed in regulating fast MyHC expression in limb muscles. Recent research has shown that the expression of fast muscle genes in developing fast limb fibres requires the transcription factor Maf, the expression of which involves the operation of the L-type Ca^2+^ channel and the elevation of the intracellular Ca^2+^ level during contraction [225]. The induction of 2B MyHC expression through neural activity is due to the binding of Maf to the 2B MyHC gene promoter [225,226]. Since the nerve connected to fast muscle also regulates masticatory fibres, it would be interesting to investigate whether Maf is involved in the regulation of masticatory fibre genes, and whether a Maf binding motif exists in the promotor region of the masticatory MyHC gene.

### 5.8. Expression of Masticatory Myosin among Vertebrates

Masticatory myosin expression is uniquely confined to jaw-closing muscles, and it is not expressed in jaw-opening muscles of crocodiles [198] and dogs [197]. Jaw-opening muscles are derived from myogenic cells in the ventral portion of the first branchial arch and belong to a different allotype. Masticatory myosin expression is widespread among carnivorous vertebrates and nut-eating mammals. It is expressed in the shark [3], a cartilaginous fish belonging to the class Chondrichthyes which first evolved the jaw. This is consistent with the fact that the masticatory MyHC gene *MYH16* is very ancient [23,227]. Masticatory myosin is also expressed in the crocodile [198] and in many mammalian taxa, including eutherian and marsupial carnivores, bandicoots and possums [100,228,229], chiropterans [230], some rodents [231] and some primates [199,230,232]. The feeding habits of these animals vary widely, and the mechanical properties of their jaw-closers may well differ. Currently, we only have kinetic properties of jaw-closers in cats and dogs [36,213]. The likely possibility that properties of masticatory myosin in different species may vary, for example, that it may be faster in small animals, as other MyHCs are, has not been investigated. However, extensive heterogeneity exists in tropomyosin and troponin T isoform expression in jaw-closing muscles between carnivores, rodents, marsupials, and reptiles [233]. It has been suggested that differences in thin-filament-mediated muscle activation evolved to accommodate markedly different feeding styles that may require high force generation in some species and high speed in others. Rodents predominantly express β-Tm, and this would reduce Ca^2+^ sensitivity to permit rodents to nibble rapidly [233].

### 5.9. Phylogenetic Plasticity of Mammalian Jaw-Closing Muscles

Mammals evolved from carnivorous reptilian ancestors that expressed masticatory myosin in jaw-closing muscles. While poikilothermic reptiles with a low metabolic rate hardly need to masticate their food for fast absorption of nutrients, it is a different matter for homeothermic mammals with their high metabolic rate. There is thus a functional demand in mammals to accelerate digestion by mastication of their food. During the mammalian radiation into various ecological niches, with animals eating different types of food, the acquisition and mastication of food required evolutionary changes not only in jaw and dental anatomy, but also in myosin isoforms and the fibre types expressed. Some eutherian (carnivores, chiropterans, primates and rodents) and marsupial (dasyurids, peramelids, diprotodonts and didelphids) taxa retained masticatory myosin expression, where a high force in jaw-closers remained functionally advantageous to their lifestyle and diet [3]. However, some members of these taxa extended their myosin expression repertoire or replaced masticatory myosin with other isoforms. Several marsupial species express masticatory fibres and α-cardiac/β-slow hybrid fibres, the ratio of masticatory to hybrid fibres being correlated with the amount of vegetable matter in their diet [229]. Among primates, the baboon extends its repertoire to comprise masticatory, 2A and β-slow myosins, while the sooty mangabey expresses masticatory, 2A, α-cardiac and β-slow myosins [232,234]. Human jaw-closers express 2X, 2A, α-cardiac, β-slow and neonatal myosins [235,236,237] but no longer express masticatory myosin, there being a frameshift mutation in the masticatory MyHC gene [238]. Of considerable interest is the fact that humans continue to express MLC1_E/A_ [239], which is associated with both masticatory and jaw slow fibres of carnivores. Among carnivora, the red panda, which is no longer carnivorous, completely replaced masticatory myosin with limb fast isoforms [199], as has *Miniopterus schreibersii* among chiropterans [230].

Herbivorous animals spend much of the day grazing and chewing the cud. They have completely replaced masticatory myosin, which has a high tension cost, with the more economic cardiac isoforms, expressing β-slow MyHC in ruminants like sheep and cattle [230,240,241] and α-cardiac MyHC in non-ruminants like macropods [242]. Though macropods and ruminants are both foregut fermenters, the former do not have a four-compartment stomach like ruminants to facilitate digestion [243]. Relative to the slow fibres of ruminants, the higher speed and power associated with α-cardiac fibres in macropods ensure the rapid comminution of food into fine particles necessary for efficient fermentation without the benefit of rumination.

Mammals that feed on diets with a wide range of textures generally have jaw-closing muscles expressing several different myosin isoforms to generate fibres with a range of kinetic properties. The rat [244], hedgehog [245] and guinea pig [246] express the four limb myosin isoforms, while the red squirrel and flying squirrel express limb fast and neonatal myosins [231]. In the rabbit, α-cardiac myosin is expressed in one-third of the jaw-closer fibre population, while the rest express 2A and β-slow myosins [235,246].

The expression of myosins in jaw-closers of marsupial mammals illustrates well the adaptation of myosin expression to varying masticatory functional demands. The dasyurids, which are exclusively carnivorous [247,248], express only masticatory myosin in jaw-closers. Bandicoots and possums feed on varying proportions of insects and vegetable matter. Their jaw-closers contain a mixture of masticatory fibres and hybrid α-cardiac/β-slow fibres, and the ratio of masticatory to α-cardiac/β-slow varies with the type of diet [229]. Bandicoots are predominantly insectivorous and have a higher proportion of masticatory fibres than possums. Brushtail possums are more frugivorous than folivorous [249] and have more masticatory fibres compared to the ringtail possum, which feeds on softer vegetable matter [250]. Kangaroos are herbivorous and have completely suppressed masticatory MyHC, expressing α-cardiac and very little β-slow MyHC [242].

The extreme versatility of the jaw-closing muscle allotype in expressing such a variety of myosins in different animals poses interesting problems of how these patterns of myosin expression are achieved at the molecular level. Future work would need to investigate the molecular basis of the phylogenetic plasticity: how the network of transcription factors that specify the masticatory muscle allotype interacts with the regulatory regions of the relevant MyHC and MLC genes in different species to bring about such diverse patterns of expression.

### 5.10. Development and Regulation of Myosin Expression in Jaw-Closing Muscles in Animals Not Expressing Masticatory Myosin

The prenatal development of rat masseter muscle reveals that the patterns of MyHC expression during myogenesis are similar to those in limb muscles [121]. All primary myotubes express β-slow and embryonic/neonatal MyHCs; a small proportion of these myotubes (slow primaries) become β-slow fibres in the mature animal, while most primary myotubes (fast primaries) become fast fibres. All secondary myotubes are negative for β-slow myosin but express embryonic/neonatal MyHC (fast secondaries) and mature into fast fibres, there being no slow secondary myotubes.

Masticatory muscles of rodents are subject to neural and hormonal regulation. Reducing the functional load tends to lead to a shift in fibre types toward fatigable 2b fibres, while increasing the functional load leads to a shift toward fatigue-resistant 2a and β-slow fibres. In rat masseter muscle, a shift of 2a fibres toward 2b fibres occurs in animals on a soft diet relative to those on a hard diet, the change being associated with reduced electromyographic activity in the masseter muscle of rats on the soft diet [251,252], suggesting that the change is mediated by a change in impulse traffic. Similar changes in MyHC expression have also been observed at the mRNA level [253]. Rabbits on a soft diet also show a reduction in β-slow and 2b fibres with increased 2a fibres [254], or reduced fibres co-expressing β-slow and α-cardiac MyHCs [255]. On the contrary, increasing the daily activity of rat masseter muscle by increasing the occlusal vertical dimension leads to an increase in 2B MyHC mRNA and a decrease in 2A MyHC mRNA [256] and the corresponding proteins [257]. Increasing the functional load in rats also increases masseter β-slow MyHC mRNA, which is completely inhibited by cyclosporin A, an inhibitor of calcineurin [258]. This suggests that the calcineurin/NFAT pathway is involved here as it is in the neural regulation of β-slow MyHC in limb muscle via CLFS, which is mediated by calcineurin. Similarly, increasing the functional load in guinea pigs leads to a shift toward β-slow myosin expression, as indicated by a reduced masseter myosin ATPase activity and a reduced average thin-filament sliding velocity of masseter myosin measured by in vitro motility assays [259].

In some rodents, fibre-type distribution in jaw-closing muscles shows sexual dimorphism, as the male animals have more powerful jaw-closers than females. In mice, males have twice as many jaw-closer fibres containing 2B MyHC as females, which have twice as many fibres containing 2A MyHC as males [260]. Similarly, in rats, there are three times more 2b fibres in the masseter muscle of males compared with females [261]. In rabbit jaw-closers, males have more 2a fibres and fewer α-cardiac/β-slow fibres than females [246]. Testosterone replacement in castrated male rabbits decreases the proportion of α-cardiac/β-slow fibres and increases that of 2a fibres, confirming the role of testosterone in generating the sexual dimorphism in fibre-type distribution [262]. In newborn guinea pigs, both male and female temporalis muscles contain a fast-red isoform, presumably 2A MyHC. At puberty, the male begins to replace the fast-red isoform with a fast-white isoform, presumably 2X/2B MyHC, accompanied by a rise in testosterone levels. Castration of the male reverses this change, while testosterone treatment of the female induces a shift toward fast-white MyHC [263]. The temporalis muscle of the baboon also shows sexual dimorphism, as females have more β-slow fibres than males [232]. Sexual dimorphism in jaw muscles contrasts with limb muscles in which MyHC expression does not respond to sex hormones [264], consistent with the allotype difference between jaw and limb muscles.

In rat masseter muscle, thyroid hormone has been shown to shift the expression of 2A MyHC mRNA toward 2B MyHC mRNA, as it does in limb muscles [210]. However, histochemical analysis of rat masseter muscle reportedly failed to demonstrate a thyroid influence on fibre-type distribution [265], but the method used was inadequate for discriminating 2a from 2b fibres. Future work using immunohistochemical methods will be needed to clarify the role of thyroid hormone in MyHC regulation in jaw-closing muscles in various species.

Acute adrenergic stimulation of limb fast and slow muscles enhances twitch and tetanic force [266]. Sympathetic nerve stimulation of rabbit jaw-closing muscle has a similar effect [267]. The enhanced force would be beneficial to an animal involved in self-defense, but little is known about the effects of acute adrenergic stimulation on jaw-closing muscles in other species. However, chronic adrenergic stimulation has been shown to induce a shift toward faster MyHCs in rat masseter muscle: following a 2-week treatment with clenbuterol, a lipophilic β_2_-adrenoceptor (β_2_-AR) agonist, an increase in 2B MyHC and a decrease in 2X and 2A MyHCs at the protein [268,269] and mRNA [258] levels occur. This is a direct effect of β_2_-AR action rather than the result of changes in motor activity [268]. This shift from 2A to 2B MyHC expression is mediated by the Akt/mammalian target of the rapamycin (mTOR) pathway [270].

## 6. Laryngeal Muscles

### 6.1. Intrinsic Laryngeal Muscles and Their Functions

Intrinsic laryngeal muscles have evolved to subserve the highly specialized and complex functions of moving the vocal fold for airway protection, respiratory control, and phonation. In mammals, there are five intrinsic laryngeal muscles: thyroarytenoid (TA), lateral cricoarytenoid (LCA), interarytenoid (IA), posterior cricoarytenoid (PCA) and cricothyroid (CT). The laryngeal muscles are classically divided into (1) adductors, comprising TA, LCA and IA, which close the glottis, (2) the abductor, PCA, which opens the glottis, and (3) the vocal fold tensor, CT. The TA has a vocalis division (TA-V) located within the vocal fold which is important in phonation and an external division (TA-X) which adducts the vocal fold to adjust airway resistance to prevent rapid lung recoil during expiration. The TA-X is also involved in reflex closure of the glottis, a movement of great importance in airway protection during sneezing and coughing. The PCA lowers airway resistance during inspiration by abducting the vocal fold, whereas the CT tenses the vocal cord and thus controls the pitch of the voice. The CT is also involved in dilating the glottis during inspiration. The various functional demands on the different laryngeal muscles within species require variations in fibre-type composition to match them. Respiratory control in animals of different body mass requires changes in kinetic properties of laryngeal muscles to deal with body mass-related changes in basal metabolic rate.

### 6.2. Developmental Origins of Laryngeal Muscles

Laryngeal muscles are derived from myogenic cells from somites 1 and 2 which migrate to the branchial arches [271]. The CT is derived from the fourth branchial arch and is innervated by the superior laryngeal nerve, a branch of the vagus (cranial nerve X). The CT has the potential of expressing only myosin isoforms found in limb muscles. All the other laryngeal muscles are derived from the sixth arch and are innervated by the recurrent laryngeal nerve, a branch of the vagus (cranial nerve X) (see Table 1). These muscles belong to a different allotype which has the potential to express the superfast myosin, EO, in addition to limb muscle myosins [107,272,273,274].

There appears to have been no studies on the early myogenesis of laryngeal muscles, but histochemical analysis of postnatal rat laryngeal muscles suggests that fibre maturation is influenced by innervation and thyroid hormone: denervation impairs the development of β-slow and 2a fibres, while hypothyroidism impairs the differentiation of 2b fibres [275].

### 6.3. Laryngeal Muscle Fibre Types

While the CT in most mammals has fibre types similar to those of limb muscles, the MyHC expression repertoire of other laryngeal muscles across species includes EO, 2B, 2X, 2A, β-slow and neonatal MyHCs, but shows systematic variations between species. The expression of EO MyHC in laryngeal muscles was first described in the rabbit [272]. In rat laryngeal muscles, an apparently laryngeal-specific MyHC was identified, initially known as 2L MyHC [273], but was subsequently shown to be identical to EO MyHC [274,276,277]. The high speed of contraction of intrinsic laryngeal muscles expressing EO MyHC in rats is associated with elevated expression levels of Ca^2+^ reuptake-related proteins of the sarcoplasmic reticulum (Ca^2+^ ATPases, phospholamban and calsquestrins) in comparison to the limb muscle [278]. Neonatal MyHC has been reported to be expressed in bovine muscle [279] and human [280] laryngeal muscles, and low levels of mRNA for this MyHC and α-MyHC have been reported in human laryngeal muscle [281]. Human TA has been reported to express slow-tonic MyHC [280,282], but this has been disputed [283].

### 6.4. Variations in MyHC Expression in Laryngeal Muscles

Two types of variation in the pattern of MyHC expression are found in laryngeal muscles: variations between muscles of different function within species and variations in specific muscles between species. Within species, the adductor TA expresses faster MyHCs or contains more fast fibres than the abductor, PCA, which expresses faster MyHCs than the vocal fold tensor, CT. This has been verified in rat, rabbit, cat, baboon [284], goat [285] and human [286] laryngeal muscles.

Between species, there is a clear inverse relationship between body mass and the speed with which MyHCs are expressed in laryngeal muscles, with the expression of the fastest MyHC within the repertoire progressively dropping out as body mass increases. EO MyHC is expressed in the TA and PCA of small animals like rats [273,287,288] and rabbits [107] but not in the larger cat and baboon. The fastest MyHC expressed in cats [107] and dogs [289] is 2B, but it is not expressed in the larger baboon [107], cattle [279] and horse [290], in which the fastest MyHC is 2X. The CT expresses only MyHC isoforms and fibre types found in the limb muscles of the same species [107].

The literature on human laryngeal muscles is highly complex. Most reports show that these muscles express 2X, 2A and β-slow MyHCs [291,292,293], and mRNA analysis has confirmed that they do not express the faster 2B and EO MyHCs [281]. However, SDS-PAGE analyses have revealed an unidentified “2L” MyHC component from human laryngeal muscles [293], including the CT, but it is absent from female subjects [294]. More recent research suggests that 2B MyHC mRNA and protein are weakly expressed in some human laryngeal muscles, and immunohistochemical analysis have revealed that 2B MyHC in humans is always co-expressed with β-slow, 2A or 2X MyHCs [295]. Therefore, it seems likely that “2L” MyHC is actually 2B MyHC, and this would also explain the high velocity of some human laryngeal fibres that exceed the velocity of the fastest human limb (2x) fibres [291]. The identification of “2L” MyHC in human CT with 2B MyHC rather than EO MyHC is coherent with the notion that the CT is a distinct allotype which does not express EO MyHC. An isoform of MyHC also thought to be EO MyHC has been reported to be expressed in the laryngeal muscle of a 7-month-old infant, and this isoform is not expressed in limb muscle [296]. This isoform is also likely to be 2B MyHC, and this would be consistent with the fact that 2B MyHC is also not expressed in human limb muscles.

Human laryngeal muscles also show considerable variability in fibre-type distribution and MyHC expression patterns between individuals [294]. As myosin expression in laryngeal muscles is subject to neural regulation (see later), variations in MyHC expression in humans could arise from different patterns of use. It would be of interest to see whether myosin expression differs in singers versus ordinary people. The possible expression of EO MyHC in human laryngeal muscle [293] is of interest in this respect. Although the highest impulse frequency of human laryngeal motor units recorded during a cough is 50 Hz [297], it is possible that humans by training can produce the very high-frequency impulses to induce EO MyHC expression in laryngeal muscles. It would be interesting in future work to determine whether very high-frequency stimulation could induce the expression of EO MyHC in laryngeal muscles of animals which do not normally express this isoform.

### 6.5. Functional Significance of Variations in MyHC Expression of Laryngeal Muscles within Species

The higher speed of contraction of the adductor TA relative to the abductor PCA and the CT in the same species may represent an evolutionary adaptation to ensure the effectiveness of the protective closure of the glottis to avoid inspiration of foreign matter into the lungs. As the abductor is active during inspiration [298], to be effective in the protective closure of the glottis, the adductor needs to be faster and more powerful than the abductor.

An important feature of laryngeal muscle fibres is the prevalence of hybrid fibres containing multiple MyHC isoforms. This has been reported in rat, rabbit, and human laryngeal muscle fibres [107,289,291,292]. This feature is probably due to the delivery of multiple MyHC regulatory impulse patterns to the same motor unit. Variations in the ratio of different MyHC isoforms in laryngeal motor units can provide a mechanism for generating laryngeal motor units with a range of speeds. For example, in the TA of the rat and rabbit, all or nearly all fibres contain EO/2B MyHCs [284]. The ratio of these MyHCs probably varies at the motor unit level, so that units with increasing ratios of EO/2B MyHCs would have increasing speeds of contraction. The recruitment of motor units of increasing speeds would help to meet the increase in respiratory frequency above the basal level during physical activity. Future work needs to investigate this possibility.

### 6.6. Regulation of MyHC Expression in Laryngeal Muscles within Species

The expression of MyHCs in laryngeal muscles within species is subject to neural and thyroidal regulation as in limb muscles. The earliest report suggesting that laryngeal muscles are subject to neural regulation came from the observation of fibre-type grouping in these muscles of horses [299], which arises from the denervation and reinnervation of muscle fibres due to recurrent laryngeal neuropathy, a common condition in horses [290]. More concrete evidence that neural traffic controls laryngeal MyHC expression has been provided by the observation that the CLFS of denervated sheep PCA muscle leads to the conversion of fast fibres into β-slow fibres [300].

Depriving laryngeal muscles of neural traffic through denervation or by interfering with neuromuscular transmission leads to changes in MyHC expression. Denervation of the PCA in rats [301,302] and humans [303] leads to a slow-to-fast fibre conversion as β-slow fibres are deprived of the CLFS needed for maintaining β-slow MyHC expression, with the denervated β-slow fibres now responding to the euthyroid state by expressing fast MyHCs. Further, in the rat TA, EO and 2B MyHCs are downregulated following denervation, replaced by 2X MyHC [302], presumably because of the loss of the very high-frequency impulses that regulate EO MyHC [95] and the high-frequency, low-amount stimuli needed for the expression of 2B MyHC [189]. Laryngeal motoneurons in the rabbit can deliver 200 Hz impulses [304]. In rats, after a single injection of botulinum toxin to block neuromuscular transmission in laryngeal muscle, a shift in MyHC expression occurs from EO toward 2B and 2X during the recovery period [305], presumably due to the filtering out of the very high-frequency neural impulses suggested for the regulation of EO MyHC expression. This response is similar to the effect of botulinum toxin on EOMs described above [185]. Laryngeal muscles of horses normally express 2X, 2A and β-slow MyHCs, but some horses with no laryngoscopic evidence of recurrent laryngeal neuropathy show fibre-type grouping and a virtual elimination of 2x fibres, presumably because of the filtering out of the high-frequency, high-amount impulse patterns promoting 2X MyHC expression [189] due to subclinical recurrent laryngeal neuropathy [290].

Neural regulation of laryngeal muscle fibres has also been demonstrated by cross-innervation experiments. Slow-to-fast fibre transformation occurs in dog PCA and LCA following cross-innervation by the hypoglossal nerve [306]. In rats, the TA-X is normally composed solely of eo/2b fibres, while the TA-V and other muscles innervated by the recurrent laryngeal nerve contain predominantly 2x fibres. Following recurrent laryngeal nerve section and reinnervation of laryngeal muscles, there was a progressive transformation of 20% of TA-X eo/2b fibres into pure 2x fibres, suggesting that nerve fibres normally innervating 2x fibres cross-innervated eo/2b fibres of the TA-X and transformed them into 2x fibres [288].

The fibre types of the rat sternohyoid (SH), a somitic muscle, comprise β-slow, 2a, 2x and 2b fibres, whereas the TA-X has only 2b/eo fibres. Following cross-innervation of these muscles, the SH retained β-slow and 2a fibres, greatly increased the proportion of the faster 2x and 2b fibres but failed to express EO MyHC. In the cross-innervated TA, the SH nerve failed to induce β-slow and 2A MyHC expression and failed to suppress EO MyHC expression in 2b/eo fibres. However, 2x fibres amounting to 4.2% appeared de novo in the TA-X [106]. These results show that the repertoires for MyHC expression of the SH and TA are determined by their muscle allotype.

Although pure eo fibres are virtually absent in the rat TA, the rabbit TA-V contains 19% of these fibres [107]. The existence of such a high proportion of pure eo fibres shows that EO MyHC is not always co-regulated with 2B MyHC, but that a distinct signal is necessary, presumably the very high-frequency neural impulses suggested for the regulation of EO MyHC in EOM [95].

Thyroid hormone also has an influence on fast MyHC expression in laryngeal muscles similar to that in limb muscles: a shift toward faster isoforms. In the adult rat TA, which expresses EO, 2B and 2X but not 2A MyHC, hyperthyroidism shifts 2X MyHC toward 2B MyHC without influencing the level of EO MyHC, while in the PCA, hyperthyroidism shifts 2A and 2X MyHCs toward 2B MyHC. Hypothyroidism has the opposite effects [287]. The expression of EO MyHC is slightly increased under hyperthyroidism in the rat TA relative to that in hypothyroid animals [287]. The functional significance of the thyroid-induced shift to a faster MyHC profile is that it helps to adjust respiratory function to match the enhanced metabolic effects of the hormone.

### 6.7. Functional Significance of Variations in MyHC Expression in Laryngeal Muscles between Species

The inverse relationship between body size and the speed with which the MyHC is expressed in laryngeal muscles serves to adjust inspiratory and expiratory airway resistances to match changes in respiratory frequency associated with changes in metabolic rate with body mass. The basal metabolic rate of eutherian and marsupial mammals is scaled approximately to (body mass)^0.75^ [307,308,309,310]. McMahon, arguing theoretically based on elastic similarity, concluded that metabolic rate should scale to (body mass)^0.75^ and physiological cycles should scale to (body mass)^−0.25^ [311]. To deliver sufficient oxygen to support of the metabolic rate, the lung ventilation rate must at least be similarly scaled to body mass. The ventilation rate, the product of tidal volume and respiratory frequency, scales to (body mass)^0.80^ [312]; the allometric exponent slightly exceeds that for basal metabolic rate, thereby ensuring adequate ventilation to support the metabolic rate as body mass changes. The respiratory frequency required to support this ventilation rate is observed to be scaled to (body mass)^−0.26^ [312], closely conforming to McMahon’s hypothesis. This means that small animals have high respiration rates and consequently less time within which to regulate airflow during each breath. The laryngeal muscles in small animals should therefore contract faster than those of large animals. The speeds of limb and cardiac MyHC isoforms do increase with decreasing body mass but are optimized for locomotion and cardiac function, respectively; the rate of change is thus inadequate to match respiratory requirements. The negative exponents of the allometric equations relating the speeds of 2b and β-slow fibres to body mass (0.041 to 0.175) [75] fall far short of those for a respiratory frequency of 0.26. To meet respiratory control requirements over the body mass range, shifts toward slower fibres in large animals and faster fibres in small animals are needed. Thus, large animals like horses [290] and baboons [107] reduce laryngeal muscle speed by expressing predominantly β-slow and 2A MyHCs, while animals at the lower end of the size spectrum need to resort to recruiting faster MyHCs not expressed in their limb muscles: 2B MyHC in cats [107] and dogs [289] and EO MyHC in rats [288]. The contractile properties of laryngeal muscles across the species reflect these fibre-type changes [96,313]. At the extremely low end of the body mass spectrum, β-slow and 2a fibres would be expected to disappear from laryngeal muscles, as they have in limb muscles of the shrew [314], while pure eo fibres would probably predominate in all laryngeal muscles of the shrew, exception perhaps in the allotypically different CT.

### 6.8. Regulation of MyHC Expression in Laryngeal Muscles between Species

Variations in MyHC expression in laryngeal muscles between species reflect changes in the properties of the cell lineage during phylogeny. The fact that the rat TA does not express 2A and β-slow MyHCs is not due to a lack of neural impulses supporting these MyHCs, since cross-innervation by a nerve supporting these MyHCs failed to induce them in the rat TA [106]. It is likely that failure to express EO MyHC in cats and dogs and failure to express EO and 2B MyHCs in baboons and horses are also not due to the lack of appropriate neural impulse patterns needed to express these MyHCs but are properties of the cell lineages. As body mass increases and the associated basal metabolic rate decreases during phylogeny, the molecular mechanisms for expressing fast MyHCs presumably become progressively less used and may eventually be lost through gene mutations, as in the case of the 2B MyHC gene in horses [315], while mechanisms regulating the slower MyHC are induced. It remains possible that disused mechanisms for expressing fast MyHCs may survive in some large animals. It would be interesting to determine whether the TA of the cat, which is near the lower end of the size spectrum that normally does not express EO MyHC, would respond to very high-frequency stimulation by expressing EO MyHC.

## 7. Overview of Myosin Expression Mechanisms in Craniofacial Muscles

This review has highlighted the fact that the pattern of myosin expression in mammalian craniofacial muscles is principally influenced by the complex interplay of cell lineages, neural impulse patterns, thyroid and other hormones, functional demands and body mass. Even before myogenesis, myoblast progenitors of craniofacial muscles from different regions of the cranial mesoderm are channeled into specific progenitor lineages with different repertoires for MyHC expression during subsequent myogenesis. During myogenesis, embryonic and fetal myoblast lineages in craniofacial muscles form primary and secondary myotubes and express the same developmental MyHCs, apparently in the same way as in limb muscles, but with very different patterns of gene expression in most craniofacial muscles during subsequent development. The various types of myotubes probably represent distinct ontotypes that determine their specific modes of response to neural and thyroid hormone influences, like the ontotypes in limb muscle.

Neural impulse patterns regulate MyHC expression in craniofacial muscles as in somitic muscles, enabling them to adjust their properties to changing functional demands during the lifetime of the animal. CLFS, which leads to the expression of β-slow MyHC in susceptible limb muscle allotypes, also induces β-slow MyHC expression in masticatory fibres of jaw muscles [220] and in sheep laryngeal muscle [300]. The different impulse patterns for the regulation of specific fast MyHCs in limb muscles probably also apply to craniofacial muscles competent in expressing these MyHCs. The much more complex expression of the numerous MyHCs of EOMs is matched by the rich diversity of impulse patterns of ocular motoneurons. It is suggested above that during EOM myogenesis, different myotube ontotypes specify the impulse patterns necessary for controlling the specific MyHC that they are destined to express. It is postulated that they do this by supplying specific neurotrophins to their innervating motoneuron via retrograde axoplasmic transport to specify the appropriate synaptic inputs to generate a variety of impulse patterns to induce different MyHC isoforms. This raises the question whether similar mechanisms are involved in generating specific impulse patterns for regulating MyHC expression in limb and other craniofacial muscles.

Thyroid hormone promotes fast limb muscle development by suppressing embryonic and neonatal MyHCs and inducing the expression of fast MyHCs in fast primary myotubes and shifts the expression of 2A and 2X MyHCs in mature fibres toward 2B MyHC, thereby enhancing muscle power. This enables the mature animal to live an active life, and the resulting higher energy expenditure is promoted by the elevated cardiac output and raised metabolic rate induced by the hormone. In rat masseter and laryngeal muscles, thyroid hormone also shifts the expression of 2A and 2X MyHCs toward 2B MyHC just as it does in limb fast muscles. The thyroid hormone-induced shift in MyHC expression toward faster isoforms in laryngeal muscles helps to adjust the respiratory rate to the higher metabolic rate induced by the hormone. In jaw-closing muscles of carnivores, thyroid hormone may also be involved in the developmental transition from embryonic and neonatal MyHC expression to that of masticatory MyHC, but this has yet to be confirmed. Much future research will be needed to determine whether thyroid hormone is also involved in the regulation of MyHC expression in EOMs during postnatal development.

Functional demands on craniofacial muscles across species vary considerably, the most complex being the demands on EOMs. To deal with demands to generate linearized saccades, smooth pursuit movements and steady fixations, EOMs expresses 11 MyHC isoforms ranging from the superfast EO MyHC to the slowest nmMyH IIB, with variations in MyHC expression along the length in some fibre types. In addition, EOM evolved specialized global and orbital layers, multiple innervation and palisade endings. Although the functional demands of EOM are highly complex, they are quite similar across species, and there is relatively little variation in fibre types and in the repertoire of MyHCs expressed in EOM fibres across species.

The functional demands on jaw-closing muscles across different mammalian species vary most widely, as mammals in different ecological niches need to adapt to the various methods of procuring and masticating a wide variety of foods. Mammals inherited the high-force masticatory MyHC from their synapsid ancestors, but many taxa have replaced it with one or more of the cardiac and limb MyHCs with properties more appropriate for the acquisition and mastication of their food. Jaw-closing muscles across species have the most varied pattern of MyHC expression, with an overall repertoire that includes the jaw-closer-specific masticatory MyHC and all skeletal and cardiac MyHC isoforms.

A functional demand of laryngeal muscles is to meet the changing respiratory requirements as the basal metabolic rate increases with decreases in body mass. Laryngeal muscles express limb MyHC isoforms and deal with this functional demand by shifting the MyHC expression profile toward slower isoforms in large animals and toward faster isoforms in small animals, even by expressing fast MyHCs not expressed in their limb muscles. Airway protection in small animals demands a very fast laryngeal adductor and is met by extending the repertoire of MyHC expression to include EO MyHC.

## 8. Future Directions

Suggestions for future work on specific topics of interest in craniofacial muscles have been made passim in this review. Although the pattern of MyHC expression in different muscle allotypes and the variations in MyHC expression among animal species are reasonably well established, we know relatively little about the molecular mechanisms involved. One issue is whether the molecular mechanism regulating a specific MyHC in craniofacial muscle is the same as that in limb muscle. The molecular basis for the neural regulation of β-slow MyHCs via CLFS is apparently the same in jaw and laryngeal muscles as in limb muscle, and probably likewise for neural impulse patterns regulating fast MyHCs in craniofacial muscles. Future work is necessary to confirm these possibilities. Recently, the deletion of the large Maf (Mafa, Mafb and Maf) transcription factor family in mice has identified these factors as major regulators for the expression of 2B MyHCs in limb muscles [226]. It would be important to determine whether ablation of large Maf, which blocks the expression of 2B MyHCs in limb muscles, would also block 2B MyHC expression in craniofacial muscles, and if so, what upstream factors in the myogenic cells of the various craniofacial allotypes turn on these Maf factors to induce 2B MyHCs.

We also do not know the molecular mechanisms that enable the jaw-closing muscle cell lineage to express limb MyHC isoforms in rats, α-cardiac MyHCs in kangaroos and β-slow MyHCs in ruminants. The promoter region of the α-cardiac MyHC in kangaroos, for example, must directly or indirectly respond to some jaw-specific factor to induce the α-cardiac MyHC in their jaw-closing muscles. Much work needs to be done to elucidate the *cis*-acting elements and *trans*-acting factors involved in the expression of the different MyHC genes in craniofacial muscles of different animal species.

Future work needs to examine the role of MyBP-C in modulating contractile properties of craniofacial muscles. Although sMyBP-C is known to be expressed in all human EOM fibres [93], its function is currently unknown. Based on studies on limb muscle [92], it is conceivable that in EOMs, the binding of the phosphorylated N-terminal of sMyBP-C to thick filaments in gSIF would enhance the eyeball rotation force at the beginning of a saccade, while the binding of unphosphorylated sMyBP-C to thin filaments in the end segments of oMIF and gMIF would increase their stiffness and help to break eyeball rotation toward the end of a saccade. This would require the expression of an appropriate protein kinase to phosphorylate sMyBP-C in gSIF and possibly a phosphatase in oMIF and gMIF. In masticatory fibres, the N-terminals of mMyBP-C may weaken the IHM of masticatory myosins along thick filaments to bring about the observed swinging out of cross-bridges toward thin filaments [44].

Relatively little work has been done on the effect of adrenergic stimulation on the mechanical properties of craniofacial muscles besides that of Grassi et al. on force enhancement in rabbit masseter muscle [267]. The enhanced force of masseter muscle following adrenergic stimulation may help to deliver an advantage in self-defense. In laryngeal muscles, β-adrenergic stimulation may result in force enhancement and an accelerated relaxation rate as in limb muscles. This would be beneficial to an animal during a fight or flight situation by facilitating an enhanced respiratory rate.

The mechanism of action of adrenaline in muscle mechanics may involve MyBP-C. As β-adrenergic stimulation activates the cAMP-activated protein kinase, which can phosphorylate sMyBP-C [92,316], this kinase is expected to phosphorylate sMyBP-C present in slow as well as in fast limb muscles following β-adrenergic stimulation. Phosphorylation of sMyBP-C would lead to the enhanced muscle force and accelerated rate of relaxation [92], just as observed following β-adrenergic stimulation in fast and slow muscles [266]. It is thus highly likely that phosphorylation of sMyBP-C underlies the molecular basis for the mechanical response to β-adrenergic stimulation in craniofacial and limb muscles [317], but this needs to be confirmed in future research. MyBP-C isoforms expressed in laryngeal muscles have not even been investigated, though it is likely that sMyBP-C is expressed in these fibres as it is in fast and slow limb fibres as well as EOM fibres. Future work on the mechanical effects of adrenergic stimulation on EOMs, jaw and laryngeal muscles and their molecular bases would be of considerable interest.

## Figures and Tables

**Table 1 ijms-25-04546-t001:** The sources of myogenic progenitor cells of extraocular, jaw and laryngeal muscles, the transcription factors involved in the specification of the various types of muscles and the nerve supplies of these muscles.

**EXTRAOCULAR MUSCLES**			
**Source of Myogenic** **Progenitor Cells**	**Transcription** **Factors**	**Muscle**	**Nerve Supply**
Prechordal plate mesoderm	Pitx2	Medial rectus Inferior rectus Superior rectus Inferior oblique	Oculomotor nerve (cranial nerve III)
Cranial paraxial mesoderm (1st branchial arch)	Pitx2 Tbx1	Superior oblique	Trochlear nerve (cranial nerve IV)
	Pitx2 Tbx1	Lateral rectus	Abducens nerve (cranial nerve VI)
**JAW MUSCLES**			
**Source of Myogenic** **Progenitor Cells**	**Transcription** **Factors**	**Muscle**	**Nerve Supply**
**Cranial** paraxial mesoderm (dorsal part of 1st branchial arch)	Pitx2 Tbx1 Capsulin/MyoR	Temporalis Masseter Medial pterygoid Lateral pterygoid (superior head)	Mandibular branch of the trigeminal nerve (cranial nerve V)
**Lateral** splanchnic mesoderm (ventral part of 1st branchial arch)	Pitx2 Tbx1	Mylohyoid Anterior digastric Lateral pterygoid (inferior head)	Mandibular branch of the trigeminal nerve (cranial nerve V)
**LARYNGEAL MUSCLES**			
**Source of Myogenic** **Progenitor Cells**	**Transcription** **Factor**	**Muscle**	**Nerve Supply**
Rostral somitic mesoderm (4th branchial arch)	Tbx1	Cricothyroid	Superior laryngeal nerve (branch of the vagus/cranial nerve X)
Rostral somitic mesoderm (6th branchial arch)	Tbx1	Thyroarytenoid Interarytenoid Lateral cricoarytenoid Posterior cricoarytenoid	Recurrent laryngeal nerve (branch of the vagus/cranial nerve X)

**Table 2 ijms-25-04546-t002:** Myosin heavy-chain expression repertoires of extraocular, masticatory, laryngeal and limb muscle allotypes across various species. Laryngeal muscles are allotypically heterogeneous; the cricothyroid and limb muscles are allotypically alike and have the same repertoire which differs from that of all the other laryngeal muscles. The myosin heavy-chain isoforms are listed approximately in the order of decreasing speed. An ‘X’ indicates that the isoform in question is expressed in adult muscles of the allotype in some species. Within an allotype, the isoform(s) expressed can vary between muscles and between homologous muscles in different species.

Myosin Heavy-Chain Isoform	Extraocular Allotype	Masticatory Allotype	Laryngeal Allotype	Cricothyroid and Limb Allotype
EO	X		X	
2B	X	X	X	X
2X	X	X	X	X
2A	X	X	X	X
Masticatory		X		
α-cardiac	X	X		
β-slow	X	X	X	X
Neonatal	X	X	X	
Embryonic	X			
Slow B	X			
Slow tonic	X			
nmMyH IIB	X			

**Table 3 ijms-25-04546-t003:** Myotube types, fibre types and MyHC composition in rabbit EOMs. The fibre types are classified as in [137], the MyHC composition of gSIFs are based on [95], and those of oSIF, oMIF and gMIF are based on [144] and assuming that slow B (MYH15) MyHC [21] and nmMyH IIB MyHC [17] are expressed in rabbit EOMs. MyHC components are listed in the order of decreasing speed; minor and variable components are in parentheses.

Myotube Type	Fibre Type	MyHC Composition
Primary	oMIF Central segment End segments	α-cardiac α-cardiac, β-slow, emb, slow B and slow tonic
Secondary	oSIF Central segment End segments	EO 2A, emb, slow B and (neo)
Primary	gMIF Central segment End segments	α-cardiac and β-slow, α-cardiac, β-slow, emb, slow tonic and nmMyH IIB
Secondary	gSIF-red	(2X), 2A and (neo)
Secondary	gSIF-intermediate	(2B), 2X and (2A and neo)
Secondary	gSIF-white	EO, 2B and (2X)

**Table 4 ijms-25-04546-t004:** MyHC expression in the four types of myotubes during myogenesis of the cat masseter muscle [204]. The brackets indicate that masticatory MyHC is transiently expressed in slow secondary myotubes.

Myotube Type	MyHCs in Myotube	MyHC in Mature Fibre
Slow primary	Emb and β-slow,	β-slow
Masticatory primary	Emb, Neo and β-slow	Masticatory
Slow secondary	Emb, Neo and β-slow, (Masticatory)	β-slow
Masticatory secondary	Emb and Neo	Masticatory

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
