# Peer review of "Developmental, Physiological and Phylogenetic Perspectives on the Expression and Regulation of Myosin Heavy Chains in Craniofacial Muscles"

_ijms, 2024, doi:10.3390/ijms25084546_

Round 1

Reviewer 1 Report

Comments and Suggestions for Authors

Major revision

The review is mainly involved in there types of craniofacial muscles, extraocular, jaw and laryngeal muscles, and systematically summarizes the progress of craniofacial muscles from developmental origin, the expression, regulation and functional significance. The review is valuable to understand the status of research on craniofacial muscles, and provide the information to investigate the mechanism of craniofacial muscles development in the future. However, the structure of the manuscript is not clear at present. The author summarized the progress for these muscles, respectively, from the origin, ontogeny, regulation, functional demands, fibre types, but the structure of the review for each muscle is different, making it very difficult for the reader to understand. In the section of abstract, the significance of this review is not well stated.

Minor revision

Line 16, MyHC IIB is a fast-twitch muslce fiber, the author describe it as the slowest, non-muscle MyHC IIB.

Line 22, it need to use the full name of EOMs where it is first appeared.

Line 24, 2A MyHC change to MyHC 2A.

The name of gene needs to be in italics, it is not appeared through the whole manuscript.

The language needs to be further optimized to make the reader to understand easily.

Comments on the Quality of English Language

The language needs to be further optimized to make the reader to understand easily.

Reviewer 2 Report

Comments and Suggestions for Authors

This is a very interesting, comprehensive review on  myosin heavy chains in  craniofacial muscles.

However, I see several issues that should be addressed before making a final decision to accept this manuscript.

1. A title is misleading. The authors concentrate only on conventional myosins and it known that several unconventional myosins are expressed and function in the muscle and myogenic cells, including myosin VI and myosin XVIII. Therefore, I suggest to add "conventional" before myosins. And of course, a paragraph mentioning the presence of unconventional myosins in the muscle should be added.

2. genes should be written in italic

3. For a reader who is not involved in myosin research, information on myosin light chains is not clear - in line 96 there are two, MLC1 and MLC2, a and 4 lines later - about three. It should be explained more clearly at the very begininng what and why...

4. It is very hard to read this review without the figures/schemes, I see a need for at least three. One could present developmental aspects of craniofacial muscle, the second could present conventional myosin heavy chain structure and association with light cains as well as myosin alignment within the filament. And third, presentation of sequence comparison within myosin heads (most important parts) of the described myosins and the light chains that could be behind the observed differences in kinetic parameters.  

Reviewer 3 Report

Comments and Suggestions for Authors

This is an extensive and extremely detailed review of craniofacial muscles with a long list of references. The author has a long-lasting experience in the study of craniofacial muscles. The review includes both descriptive analyses and discussion of mechanisms and open issues. This will be a precious resource to all working in this field.

General points

1. I would suggest reorganizing the various sections this way:

1. Introduction

2. Myosin in Craniofacial Muscles   2.1 ... 2.2 … etc.     

3. Craniofacial Muscle Allotypes

4. Extraocular Muscles  3.1 … 3.2 …  etc.

5. Jaw Muscles   4.1 … 4.2 … etc.

6. Laryngeal Muscles  5.1 …  5.2 ….  etc.  

7. Overview of Myosin Expression Mechanisms in Craniofacial Muscles

8. Future directions

2. The author might consider complementing the Introduction with some general considerations not only on ontogeny but also on phylogeny of craniofacial muscles (a possible title could be: “Craniofacial muscles: an introduction to their ontogeny and phylogeny”). Here, one could stress the point that some of these muscles, such as the EOMs, are present in early vertebrates, such as the lamprey, and show the same anatomical pattern (4 recti and 2 oblique muscles) like in all other vertebrate classes. In contrast, other craniofacial muscles, such as the jaw muscles, which are not present in jawless vertebrates, display wide variations among species …

Specific points

Use italics throughout whenever you refer to genes (e.g.  line 115-6: encoded by MYL4 and MYL3)

Line 21-23: “The primary and secondary myotubes of developing EOMs are postulated to induce, via different retrogradely transported neurotrophins, the rich diversity of neural impulse patterns that regulate the specific MyHCs they express.” This has not been demonstrated.  Several lines of evidence rather support the notion that diversification into different cell types occurs independently in muscle cells and motor neurons and appropriate matching occurs following a period of polyneuronal innervation (neonatal period in rat limb muscles).  The independent diversification of muscle cells and motor neurons has been clearly demonstrated in both mammalian and avian systems (see Hughes & Salinas, Curr Opin Neurobiol 1999, and Landmesser, Int J Dev Neurosci 2001).

Line 126: motif

Line 128: enhanced

Line 134: add at the end of the sentence: although definite evidence based on in vitro motility assays is still missing (see Schiaffino et al, J Physiol 2024)

Line 150 & 152: Slow B myosin… Change to:  MYH15 myosin

Line 151: amphibian and avian ventricular MyHC …

Line 324-5: You should add that according to Rossi et al (2010) slow-tonic MYH is expressed in all fibres in fetal EO muscles and disappears in most fibres, except the slow-tonic fibres, during early postnatal stages.

Line 337: use MYH15 instead of slow B

Line 583-4: “… chronic low frequency stimulation (CLFS) controls beta-slow MyHC expression in this muscle and can convert other muscle fibre types into beta-slow fibres (see later).” The author should better specify that the chronic electrical stimulation studies in rat limb muscles clearly showed that it is difficult to induce slow myosin in denervated EDL or 2B myosin in soleus, consistent with the notion that intrinsic differences between fast and slow muscles dictate the “adaptive ranges” of possible fiber transformations induced by stimulation (Westgaard and Lomo, J Neurosci 1988), a notion supported by myosin analyses (Ausoni et al., ref. 184). Similar limitations in fiber type conversion are seen in response to changes in thyroid hormone, as you correctly point out later (line 696-8).

Line  679-82: This point should be better specified, as it was shown that in rat hindlimb muscles 2X-, 2A-, and 2B-MyHC transcripts are first detected by postnatal day 2-5 and display from the earliest stages a distinct pattern of distribution in different muscles and different fibers, which is similar to that seen in the adult (DeNardi et al, JCB 1993).

Line 756-7: “It is likely that the molecular mechanism involved here is the same as that which transform limb fast fibres into beta-slow fibres”.  See above, line 583.

Line 779: The fact that masticatory myosin is expressed in sharks is based on a review (ref. 90) in which this finding is referred to as unpublished observation. Was a paper ever published? The authors could point out that Desjardins et al (ref 13) showed that MYH16 is one of the most ancient myosins in vertebrate evolution. Another reference may be appropriate: Lee … Leinwand, Skelet Muscle 9:7, 2019.  

Line 991-2: any ref. about baritones and sopranos? If not, better say “It would be of interest to see whether myosin expression differs in singers versus …)

Line 1141 & line 1194:  See comment on line 583 above.

Line 1146-50: This is not correct, see comment on line 21.

Line 1154-58: You should better explain how the shift from the more oxidative 2A and 2X towards the less oxidative 2B fits with the higher metabolic rate induced by the thyroid hormone.

Round 2

Reviewer 1 Report

Comments and Suggestions for Authors

I have to say that the author provide a pretty comprehensive review which is helpful to understand the knowledge of Craniofacial Muscles, and also revised the structure of the manuscript according to the suggestion. I have no other comments to the review, except that I don't think the review highlights the significance very well. What is the ultimate purpose and significance of understanding the regulatory mechanisms of Craniofacial Muscles

Author Response

            The ultimate purpose and significance of understanding the regulatory mechanisms of craniofacial muscles is to understand how different types of muscle work. The review shows that regulatory mechanisms for myosin expression in craniofacial muscles are very similar to those of limb muscles in that they are influenced by the complex interplay of cell lineages, neural impulse patterns, thyroid and other hormones, functional demands and body mass. However, they differ radically in the type of cell lineage and the nature of their functional demands. I have added a sentence in the abstract to point this out:

"In these respects, craniofacial muscles are similar to limb muscles, but they differ radically in the types of cell lineage and the nature of their functional demands.”

"

Reviewer 2 Report

Comments and Suggestions for Authors

The authors replied to all my comments.

Author Response

The reviewer seems to be satisfied with my replies to all his comments.